# Mining Unseen Classes via Regional Objectness: A Simple Baseline for Incremental Segmentation

Zekang Zhang[1],[*]   Guangyu Gao[1],[†]   Zhiyuan Fang[1],   Jianbo Jiao[2],   Yunchao Wei[3],[4]

[1] School of Computer Science, Beijing Institute of Technology
[2] School of Computer Science, University of Birmingham
[3] WEI Lab, Institute of Information Science, Beijing Jiaotong University
[4] Beijing Key Laboratory of Advanced Information Science and Network
zkzhang1998@outlook.com

## Abstract

Incremental or continual learning has been extensively studied for image classification tasks to alleviate *catastrophic forgetting*, a phenomenon that earlier learned knowledge is forgotten when learning new concepts. For class incremental semantic segmentation, such a phenomenon often becomes much worse due to the background shift, *i.e.*, some concepts learned at previous stages are assigned to the background class at the current training stage, therefore, significantly reducing the performance of these old concepts. To address this issue, we propose a simple yet effective method in this paper, named **Mi**ning unseen **C**lasses via **R**egional **O**bjectness for **Seg**mentation (MicroSeg). Our MicroSeg is based on the assumption that *background regions with strong objectness possibly belong to those concepts in the historical or future stages*. Therefore, to avoid forgetting old knowledge at the current training stage, our MicroSeg first splits the given image into hundreds of segment proposals with a proposal generator. Those segment proposals with strong objectness from the background are then clustered and assigned newly-defined labels during the optimization. In this way, the distribution characterizes of old concepts in the feature space could be better perceived, relieving the *catastrophic forgetting* caused by the background shift accordingly. Extensive experiments on Pascal VOC and ADE20K datasets show competitive results with state-of-the-art, well validating the effectiveness of the proposed MicroSeg. Code is available at https://github.com/zkzhang98/MicroSeg.

## 1 Introduction

Deep learning based methods have made significant achievements on various recognition tasks, while assuming the data distribution to be fixed or stable and the training samples are independently and identically distributed [18]. However, in practical scenarios, the data always presents a continuous stream without a stable distribution. Under these scenarios, the old knowledge will be easily interfered with or even totally forgotten by continuously acquiring new knowledge, which is generally considered to be *catastrophic forgetting* [3, 24, 28]. To tackle this problem, many approaches were proposed to incrementally learn new concepts over time but without forgetting old knowledge, especially for the classification task, *i.e.*, Class-Incremental Learning (CIL) [17, 19].

Meanwhile, Class-Incremental Semantic Segmentation (CISS), aiming to predict a pixel-wise mask for an image in CIL scenarios, is crucial for applications like autonomous driving and robotics.

---

[*]Work done during an internship at WEI Lab of Beijing Jiaotong University.
[†]Corresponding author.

36th Conference on Neural Information Processing Systems (NeurIPS 2022).

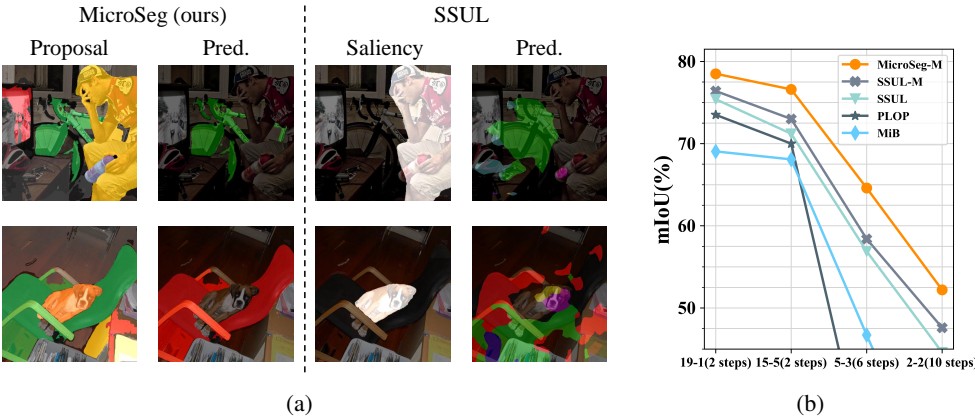

(a)                                                    (b)

Figure 1: (a) Comparison of our proposed MicroSeg and the state-of-the-art SSUL [5]. Saliency detection only points out the most important area of an image, leaving the rest unseen classes result in semantic shift and lead to incorrect predictions. The proposed proposal-based approach attends each pixels within the sample, including both *things* and *stuff*. (b) Performance comparison of prior works when the number of incremental step increases (mIoU < 45% has been dropped out).

Nevertheless, besides *catastrophic forgetting of old concepts* in CIL, CISS involves an additional key issue of *background shift* in the image [4], making CISS rather challenging. Specifically, *background shift* means that the class of background (all other classes that are not learned at the moment) is shifting over the learning steps. As a result, such background shift will confuse the model and make it more difficult to distinguish different unseen classes, which generally include the true background (dummy) class, the historical classes, and future classes.

The problems of catastrophic forgetting and background shift in CISS have been studied in recent works [4, 5, 11]. MiB [4] proposed a distillation-based framework to address the background shift issue. PLOP [11] re-models the background labels with pseudo-labeling w.r.t. predictions of the old model for historical classes. However, these prior works did not particularly address the future classes issue that leads to background shift, while instead treating those future classes as background. These future classes will not only interfere with old knowledge due to catastrophic forgetting, but also be constantly introduced as current classes to be accurately identified. The state-of-the-art method SSUL [5] tried to address the future class issue by introducing an unknown class according to saliency detection. However, as shown in Fig. 1 (a), as saliency only highlights a few regions within an image, most unseen classes still remain in the background and result in background shift. As shown in Fig. 1 (b), with the number of incremental steps increasing, the performance of the aforementioned methods has a dramatic performance drop, suggesting that the background shift issue still apparently exists, especially for the challenging setting of more learning steps. Whereas for our approach (MicroSeg-M in Fig. 1 (b)), it outperforms prior works at each step. Furthermore, if considering the incremental learner lies in a lifelong learning process, the number of unseen classes would be infinite and diverse with endless learning steps. Thus, the unseen classes in each learning step become even more non-negligible, which interfere not only with maintaining concepts of historical classes, but also with the learning of current classes.

To address the above-mentioned issues in CISS, especially the unseen classes, we propose a simple yet effective approach for **Mi**ning unseen **C**lasses via **R**egional **O**bjectness for the **Seg**mentation task (MicroSeg). MicroSeg first generates a set of segment proposals for input samples with a proposal generator. Each proposal is a binary mask pointing out a segment with strong objectness in an image. This mask is also class-agnostic and can generalize even to unseen categories. With the assumption of *background regions with strong objectness belong to concepts either in the historical or future steps*, the proposals from the background can be confidently clustered and re-modeled as *unseen classes*. Following that, proposal-based mask classification [8, 9] is applied to label each proposal with current classes and unseen classes. Moreover, to better cache the feature diversity in loose future classes into particular unseen classes, multiple sub-classes are explicitly defined and clustered according to a contrastive loss. As shown in Fig. 1 (a), the background shift issue gets alleviated with the discovery of unseen classes, showing the effectiveness of the proposed MicroSeg in reducing semantic ambiguity within categories.

Extensive experimental analysis on Pascal VOC2012 and ADE20K show the effectiveness and competitive performance of the proposed MicroSeg. MicroSeg consistently achieves higher performance across all benchmark datasets, on all incremental learning settings, *e.g.*, an average of $3\%$ improvements over the state-of-the-art method on Pascal VOC2012 and $2\%$ on ADE20K. Particularly, MicroSeg outperforms prior works in the more challenging setting of long-term incremental scenarios, *e.g.*, the performance gain of **6.2%** on Pascal VOC 2012.

## 2 Related work

**Class incremental learning (CIL).** Class incremental learning (CIL) is incremental learning bound to classification tasks. Due to the catastrophic forgetting, some approaches consider CIL as a *trade-off* between defying forgetting and learning from new samples of a network. The earlier work to deal with catastrophic forgetting is mainly about parameter isolation [2, 23, 33, 34, 35, 40], which assigns isolated models parameters for each task, while the number of parameters will always grow without an upper bound with task increases. Replay-based approaches [10, 22, 30, 31, 36] were proposed by maintaining a sampler of data for future training. Later, regularization-based methods [1, 17, 19, 29, 41], such as knowledge distillation [15] appears, which rely on a teacher model for transferring knowledge of previous categories to current models.

**Semantic segmentation.** In semantic segmentation, deep models have achieved outstanding performance. FCN [21] and U-Net [32] replace fully connected layers with convolution layers to preserve the location information of pixels. The following methods [16, 25, 38, 39, 44] are mainly based on FCN and enlarge the perceptual field, making it more suitable for semantic segmentation tasks that require intensive annotation. DeepLab series methods [6, 7] contains an Atrous Spatial Pyramid Pooling (ASPP) for semantic segmentation using dilated convolution to capture multi-scale information. Meanwhile, MaskFormer [9] regards the semantic segmentation as a mask classification task and introduces the transformer [37] to obtain a better performance. Specifically, they first generate mask proposals and then perform mask classification with the queries generated by the transformer.

**Class incremental semantic segmentation (CISS).** Previous work has made attempts at semantic segmentation in incremental learning scenarios. Modeling-the-Background (MiB) [4] first clarifies the background shift in CISS, and proposed a distillation-based framework as in CIL. The subsequent work of Sparse and Disentangled Representations (SDR) [27] exploited a latent space to reduce forgetting, especially stored prototypes (class centroids) to recall the previous categories. Meanwhile, Douillard et al. [11] proposed the approach of PLOP to utilize the pseudo-label of the background pixels in the current step with the previously learned model as supervision. Furthermore, the SSUL [5] introduces an additional unknown class label to unseen objects in the background by an off-the-shelf saliency-map detector, preventing the model from forgetting by freezing model parameters.

## 3 Method

### 3.1 Problem Definition

We follow the common definition of Class Incremental Semantic Segmentation (CISS) as in the previous work [5, 11], but also give a more clear definition on the *background class* and *unseen classes*. In CISS, there are a series of incremental *learning steps*, as $t = 1, \ldots, T$. Let $\mathcal{D}_t$ denotes the dataset for the $t_{th}$ learning step, and $\mathcal{C}_t$ the class set of samples in dataset $\mathcal{D}_t$. For any pair $(\boldsymbol{x}_t, \boldsymbol{y}_t)$ within $\mathcal{D}_t$, $\boldsymbol{x}_t = \{\boldsymbol{x}_{t,i}\}_{i=1}^{Q}$ and $\boldsymbol{y}_t = \{\boldsymbol{y}_{t,i}\}_{i=1}^{Q}$ denote the input image and its ground-truth mask with $Q$ pixels, respectively. Unlike previous approaches that specifically define a background class, we argue that in a lifelong incremental learning process with annotations of both stuff and thing, all pixels that not belong to current class will be learned in the future. Therefore, in each learning step, the class set can be expressed as $\mathcal{C}_t \cup \{c_u\}$, where $\mathcal{C}_t$ denotes the classes to be learned currently, and $c_u$ denotes a special class consisting of all the historical and future classes. $c_u$ can also be considered as the background class from the perspective of each current learning step, and varies for each learning step, where the *background shift* lies.

The CISS model $f_{t,\theta}$ with parameters $\theta$ at the $t_{th}$ learning step, aims at assigning each pixel of $\boldsymbol{x}_t$ with the probability for each class. The model $f_{t,\theta}$ includes a convolutional feature extractor and a

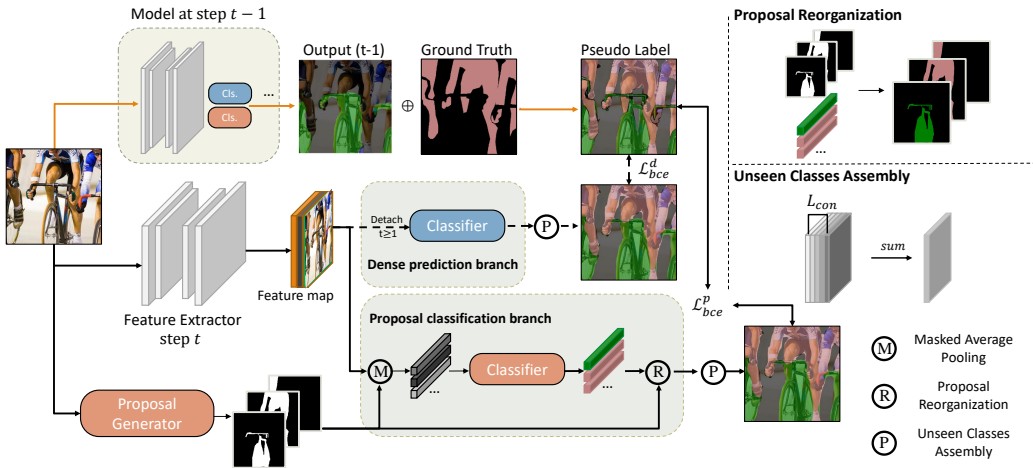

Figure 2: Overall architecture of the proposed MicroSeg.

classifier (to classify each category within the union of $\mathcal{C}_{1:t} = \bigcup_{i=1}^{t} \mathcal{C}_i$ and the unseen class $c_u$, *i.e.*, $\mathcal{C} = \mathcal{C}_{1:t} \cup c_u$ ). The model after the learning step $t$ gives a prediction for all classes, within historical classes $\mathcal{C}_{1:t-1}$. Specifically, the predicted result can be expressed as $\hat{\boldsymbol{y}}_t = \arg\max_{c \in \mathcal{C}} f^c_{t,\boldsymbol{\theta}}(\boldsymbol{x})$. Thus, catastrophic forgetting happens due to the lack of historical and future classes supervision in $\mathcal{C}_t$.

## 3.2 Overview of the MicroSeg

Fig. 2 illustrates the overall framework of our proposed method. We design a two-branch structure with the proposed Mining unseen Classes via Regional Objectness (Micro) mechanism. Given an input, the upper branch in Fig. 2 performs dense prediction over the whole image, while the lower branch splits it into segment proposals by proposal generator and classifies them into corresponding categories. In particular, we introduce the *micro mechanism* in the optimization process of both branches, to address the background shift issue.

**Proposal generator.** The proposal generator aims to generate a group of binary masks $\boldsymbol{P} \in \{0,1\}^{N \times H \times W}$ where $N$ is the number of proposals and $H, W$ denote height and width. Unlike previous semantic segmentation methods, we split the task into class-agnostic proposal generation by Mask2Former [8] and proposal classification. In this study, all proposals are *disjoint*, and each pixel belongs to and only belongs to one proposal.

**The dense prediction branch.** The CISS model $f_{t,\boldsymbol{\theta}}$ is composed of a feature extractor $g_{t,\boldsymbol{\theta}_1}$ and a classifier $h_{t,\boldsymbol{\theta}_2}$. For convenience, we abbreviate $f_{t,\boldsymbol{\theta}}$ as $f = h \circ g(\cdot)$. Note that the proposal classification branch and dense prediction branch share the same structure and parameters of $g(\cdot)$, but with different designs of $h(\cdot)$, *i.e.*, $h_p(\cdot)$ and $h_d(\cdot)$. The dense prediction branch performs conventional semantic segmentation, *i.e.*, $\boldsymbol{p}_d = h_d(g(\boldsymbol{x}))$ where $\boldsymbol{p}_d$ indicates the prediction scores (logits).

**The proposal classification branch.** In this branch, masked average pooling (MAP) [42] is used to generate prototype (centroid) $\boldsymbol{pr} \in \mathbb{R}^{N \times C}$ of each proposal, where $C$ is the number of channels of feature map. The logits $\boldsymbol{p} \in \mathbb{R}^{N \times |\mathcal{C}|}$ of all proposals are generated by classifying $\boldsymbol{pr}$, where $|\mathcal{C}| = |\mathcal{C}_{1:t} \cup \{c_u\}|$ denotes number of $\mathcal{C}$. Then $\boldsymbol{p}$ is *reorganized* to the original shape and the prediction scores are assigned accordingly. Formally, the proposal classification branch can be expressed as:

$$\boldsymbol{p}_p = Reg(h_p(MAP(g(\boldsymbol{x}), \boldsymbol{P})), \boldsymbol{P}), \tag{1}$$

where $Reg(\cdot)$ is the Proposal Reorganization shown in Fig. 2, and can be implemented by matrix multiplication, detailed in Supplementary Materials. The proposal classification branch focuses on objectness aggregation within a proposal, aiming at consistent representations in a particular proposal.

### 3.3 Mining Unseen Classes via Regional Objectness

Due to the background shift issue (*i.e.*, background class has unclear and unstable semantics during the training steps) mentioned above, the unseen class here refers to all the classes exclude the current foreground class. As mentioned in the observation in Sec. 1, this problem was not well explored in prior works, due to the ignorance of unseen classes. The proposed MicroSeg, on the other hand, can discover the unseen area with the proposal classification branch. Based on the assumption that background regions with strong objectness possibly belong to those concepts in the historical or future steps, we further remodel the supervision label and **mi**ning unseen **c**lasses via **r**egional **o**bjectness, namely, Micro mechanism. Mirco mechanism clusters proposals with strong objectness in unseen classes, and assigns them with new defined labels during the optimization.

**Label remodeling.** To capture the knowledge learned in the past learning step $t-1$, we extract the sample on $f_{t-1,\theta}$ during inference to get the pseudo label of mask $\hat{\boldsymbol{y}}_{\boldsymbol{t-1}}$ and the corresponding prediction score map $\boldsymbol{s}_{\boldsymbol{t-1}}$. Formally:

$$
\begin{aligned}
\hat{\boldsymbol{y}}_{\boldsymbol{t-1}} &= \arg \max_{\mathcal{C}_{1:t-1}} f_{t-1,\boldsymbol{\theta}}(\boldsymbol{x}), \\
\boldsymbol{s}_{\boldsymbol{t-1}} &= \max_{\mathcal{C}_{1:t-1}} \sigma(f_{t-1,\boldsymbol{\theta}}(\boldsymbol{x})),
\end{aligned}
\tag{2}
$$

where $\sigma(\cdot)$ denotes Sigmoid function. With the ground truth mask $\boldsymbol{y}_{\boldsymbol{t}} = \{\boldsymbol{y}_{t,i}\}$ in current step, we augment the supervision label $\tilde{\boldsymbol{y}}_{t,i}$ of pixel $i$ according to the following rules:

$$
\tilde{\boldsymbol{y}}_{t,i} = \begin{cases} \boldsymbol{y}_{\boldsymbol{t,i}} & \text{where } \boldsymbol{y}_{t,i} \in \mathcal{C}_t \\ \hat{\boldsymbol{y}}_{\boldsymbol{t-1,i}} & \text{where } \boldsymbol{y}_{t,i} = c_u \wedge \boldsymbol{s}_{\boldsymbol{t-1,i}} > \tau \\ c_{\hat{u}} & \text{others } ( \text{ or } \boldsymbol{y}_{t,i} = c_u \setminus \mathcal{C}_{1:t-1}) \end{cases}
\tag{3}
$$

where $\tau$ represents a threshold for $\hat{\boldsymbol{y}}_{\boldsymbol{t-1,i}}$, $c_{\hat{u}}$ denotes *future classes* within the unseen class $c_u$, the symbol $\wedge$ represents the co-taking of conditions and $\setminus$ means the difference of the sets.

**Micro mechanism.** The micro mechanism is introduced to characterize the various semantics in unseen classes during the optimization and to alleviate the background shift issue. Even the future classes $c_{\hat{u}}$ include several classes of stuff or things, so we represent it as a set of $K$ classes to better cache the feature diversity in those loose future classes. Formally, we represent $c_{\hat{u}} = \{c_{\hat{u},k}\}_{k=1}^{K}$, where each $c_{\hat{u},k}$ can be viewed as a cluster center of all the future classes. We impose both supervised and unsupervised constraints (detailed in Sec. 3.4) on $c_{\hat{u}}$ during the optimization. For the supervised part, the prediction scores of class $c_{\hat{u}}$ are assembled with the summary of the logits of $c_{\hat{u},k}$:

$$
\boldsymbol{p}_{c_{\hat{u}}} = \sum_{k=1}^{K} \boldsymbol{p}_k(\boldsymbol{x}),
\tag{4}
$$

where $\boldsymbol{p}_k(\boldsymbol{x}) = f_{t,\boldsymbol{\theta}}^{c_{\hat{u},k}}(\boldsymbol{x}) \in \mathbb{R}^{H \times W}$ is the the prediction scores of class $c_{\hat{u},k}$. $H$ and $W$ denote the height and width of image samples respectively.

We argue that $c_{\hat{u}}$ captures more features with different spatial characteristics in unseen classes. This prevents potential classes from being incorrectly classified into historical classes $\mathcal{C}_{1:t-1}$, which reduces the impact on historical knowledge and the collapse of class predictions. Besides, we equipped our method with the memory sampling strategy [5], resulting in *MicroSeg-M*. With rehearsal of samples from historical learning steps, catastrophic forgetting is alleviated significantly.

### 3.4 Objective Function

MicroSeg constrains the two branches via Binary Cross-Entropy (BCE) loss, calculated with augmented supervision label $\tilde{\boldsymbol{y}}$ from Eq. 3. Specifically, $\mathcal{L}_{BCE}$ is defined as:

$$
\begin{aligned}
\mathcal{L}_{BCE} = -\frac{1}{|\mathcal{C}|}\frac{1}{Q}\sum_{l=1}^{|\mathcal{C}|}\sum_{i=1}^{Q}(\mathbb{1}[\tilde{\boldsymbol{y}}_{t,i} = c_l]\log(\sigma(\boldsymbol{p}_l^i)) + (1 - \mathbb{1}[\tilde{\boldsymbol{y}}_{t,i} = c_l])\log(1 - \sigma(\boldsymbol{p}_l^i))) \\
-\frac{1}{Q}\sum_{i=1}^{Q}(\mathbb{1}[\tilde{\boldsymbol{y}}_{t,i} = c_{\hat{u}}]\log(\sigma(\boldsymbol{p}_{c_{\hat{u}}}^i)) + (1 - \mathbb{1}[\tilde{\boldsymbol{y}}_{t,i} = c_{\hat{u}}])\log(1 - \sigma(\boldsymbol{p}_{c_{\hat{u}}}^i))),
\end{aligned}
\tag{5}
$$

where $\boldsymbol{p_l^i}$ means the predicted logits of the $i_{th}$ pixel to the $l_{th}$ class in $\mathcal{C}$, and $\boldsymbol{p}_{c_{\hat{u}}}^i$ corresponds to $c_{\hat{u}}$ in Eq. 3. We note BCE loss of two branches as $\mathcal{L}_{BCE}^p$ and $\mathcal{L}_{BCE}^d$.

Benefiting from the class-agnostic proposals, the proposal classification is able to discover the unseen classes in CIL. Meanwhile, the dense prediction is still helpful when the supervised label is complete, *i.e.*, the first learning step. The reason is that there is no historical class at the very initial learning step, where catastrophic forgetting is not yet happened. The dense prediction branch is also beneficial to learn a robust and versatile feature extractor on base classes, which is vital for the incremental settings, where historical samples are not available in the latter learning steps. The output of the whole model is from the proposal classification branch. The loss for this part is then defined as:

$$\mathcal{L}_{BCE} = \begin{cases} \mathcal{L}_{BCE}^p + \mathcal{L}_{BCE}^d & \text{if } t = 1 \\ \mathcal{L}_{BCE}^p & \text{others} \end{cases}. \tag{6}$$

We also apply another constraint to enhance the unseen classes modeling, *i.e.*, the contrastive loss [13], to capture the diverse concepts in $c_{\hat{u}}$:

$$\mathcal{L}_c = -\frac{1}{K} \sum_{i=1}^{K} \log \frac{\exp(\boldsymbol{p_i} \cdot \boldsymbol{p_i})}{\sum_{j=1}^{K} \exp(\boldsymbol{p_i} \cdot \boldsymbol{p_j})}, \tag{7}$$

where $\cdot$ denotes the inner product. In Micro mechanism, we represent future classes $c_{\hat{u}}$ as explicitly-defined multiple sub-classes to better cache the feature diversity in those loose future classes. To avoid model degradation caused by $c_{\hat{u},k}$ characterizing the same features, we use unsupervised contrastive loss to force them to capture the diverse concepts, thus ensuring the effectiveness of the Micro mechanism.

Finally, the overall loss function is defined as below with a hyper-parameter $\lambda$ to balance the two terms:

$$\mathcal{L} = \mathcal{L}_{BCE} + \lambda \cdot \mathcal{L}_c. \tag{8}$$

# 4 Experiments

## 4.1 Experimental Setups

**Dataset.** We have evaluated our approach on datasets of Pascal VOC 2012 [12] and ADE20K [43]. Pascal VOC 2012 contains 10,582 training images and 1449 validation images, with total of 20 foreground classes and one background class. ADE20K contains 20,210 training images and 2,000 validation images with 100 thing classes and 50 stuff classes.

**Protocols.** Following the conventions of previous work [5, 11], we evaluate our approach with *overlapped* experimental setup, which is more realistic and challenging than the *disjoint* setting studied by [4, 27]. For the *incremental scenario*, we apply the same representation as in the previous works [4, 27]. For example, the 15-1 scenario of Pascal VOC 2012 (including 20 foreground classes) takes 6 steps to train the model, *i.e.*, 15 classes are used in step 1 and one class for each step in steps 2 to 6. More details of protocols are shown in the supplementary materials.

**Implementation details.** Following the common practice [4, 5, 27], we use DeepLabv3 [6] and ResNet-101 [14] pretrained on ImageNet as the segmentation network, and the output stride is 16. We train the network using the SGD with a learning rate of $10^{-2}$ on Pascal VOC2012, and $5 \times 10^{-3}$ for all steps on ADE20K. The momentum and weight decay are 0.9 and $10^{-4}$ on both datasets. The batch size is 16 for Pascal VOC 2012, and 12 for ADE20K. Data augmentation from [11] is applied to all the images. MicroSeg freezes the parameters of the feature extractor in the learning steps $t \geq 1$, following [5]. Our experiments are implemented with PyTorch on two NVIDIA GeForce RTX 3090 GPUs. We choose Mask2Former [8] pre-trained on MS-COCO [20] to generate $N = 100$ class-agnostic proposals. Note that, for fair comparisons, the proposal generator is **not** fine-tuned on any benchmark dataset. The setting of Micro mechanism is $K = 5$ for Pascal VOC 2012 and $K = 1$ for ADE20K. Thus, all pixels of samples in ADE20K can be potential foreground. Hyper-parameters $\tau = 0.7$ and $\lambda = 1$ are set for all experiments. With the same setting of memory sampling strategy as [5], we set the $|\mathcal{M}| = 100$ for Pascal VOC 2012 and $|\mathcal{M}| = 300$ for ADE20K to evaluate MicroSeg-M. More details are shown in the supplementary materials.

Table 1: Comparison with state-of-the-art methods on Pascal VOC 2012.

| Method | VOC 10-1 (11 steps) | | | VOC 2-2 (10 steps) | | | VOC 15-1 (6 steps) | | | VOC 5-3 (6 steps) | | | VOC 19-1 (2 steps) | | | VOC 15-5 (2 steps) | | |
|---|---|---|---|---|---|---|---|---|---|---|---|---|---|---|---|---|---|---|
| | 0-10 | 11-20 | all | 0-2 | 2-18 | all | 0-15 | 16-20 | all | 0-5 | 6-20 | all | 0-19 | 20 | all | 0-15 | 16-20 | all |
| LwF-MC [19] | 4.7 | 5.9 | 4.9 | 3.5 | 4.7 | 4.5 | 6.4 | 8.4 | 6.9 | 20.9 | 36.7 | 24.7 | 64.4 | 13.3 | 61.9 | 58.1 | 35.0 | 52.3 |
| ILT [26] | 7.2 | 3.7 | 5.5 | 5.8 | 5.0 | 5.1 | 8.8 | 8.0 | 8.6 | 22.5 | 31.7 | 29.0 | 67.8 | 10.9 | 65.1 | 67.1 | 39.2 | 60.5 |
| MiB [4] | 12.3 | 13.1 | 12.7 | 41.1 | 23.4 | 25.9 | 34.2 | 13.5 | 29.3 | 57.1 | 42.6 | 46.7 | 71.4 | 23.6 | 69.2 | 76.4 | 50.0 | 70.1 |
| SDR [27] | 32.1 | 17.0 | 24.9 | 13.0 | 5.1 | 6.2 | 44.7 | 21.8 | 39.2 | 12.1 | 6.5 | 8.1 | 69.1 | 32.6 | 67.4 | 57.4 | 52.6 | 69.9 |
| PLOP [11] | 44.0 | 15.5 | 30.5 | 24.1 | 11.9 | 13.7 | 65.1 | 21.1 | 54.6 | 17.5 | 19.2 | 18.7 | 75.4 | 37.4 | 73.5 | 75.7 | 51.7 | 70.1 |
| SSUL [5] | 71.3 | 46.0 | 59.3 | 62.4 | 42.5 | 45.3 | 77.3 | 36.6 | 67.6 | 72.4 | 50.7 | 56.9 | 77.7 | 29.7 | 75.4 | 77.8 | 50.1 | 71.2 |
| SSUL-M [5] | 74.0 | 53.2 | 64.1 | 58.8 | 45.8 | 47.6 | 78.4 | 49.0 | 71.4 | 71.3 | 53.2 | 58.4 | 77.8 | 49.8 | 76.5 | 78.4 | 55.8 | 73.0 |
| MicroSeg (Ours) | 72.6 | 48.7 | 61.2 | 61.4 | 40.6 | 43.5 | 80.1 | 36.8 | 69.8 | **77.6** | 59.0 | 64.3 | 78.8 | 14.0 | 75.7 | 80.4 | 52.8 | 73.8 |
| MicroSeg-M (Ours) | **77.2** | **57.2** | **67.7** | **60.0** | **50.9** | **52.2** | **81.3** | **52.5** | **74.4** | 74.8 | **60.5** | **64.6** | **79.3** | **62.9** | **78.5** | **82.0** | **59.2** | **76.6** |
| Joint | 82.1 | 79.6 | 80.9 | 76.5 | 81.6 | 80.9 | 82.7 | 75.0 | 80.9 | 81.4 | 80.7 | 80.9 | 81.0 | 79.1 | 80.9 | 82.7 | 75.0 | 80.9 |

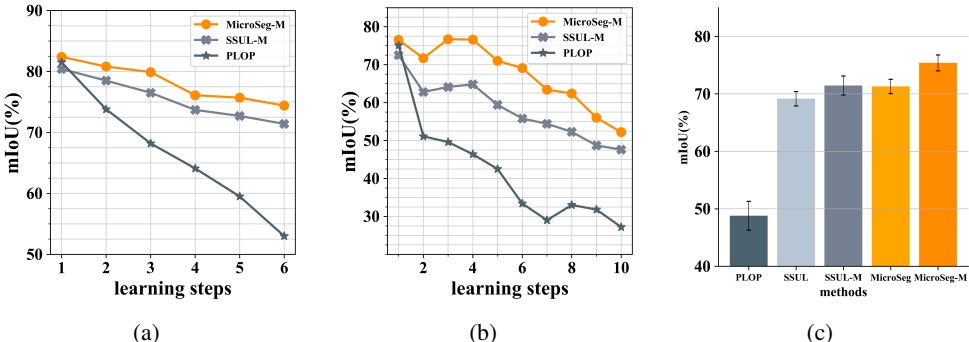

(a)        (b)        (c)

Figure 3: (a) mIoU visualization on Pascal VOC 2012 15-1, (b) mIoU visualization on Pascal VOC 2012 2-2, (c) mIoU on Pascal VOC 2012 in scenario 15-1, averaged over 20 different training orders.

**Baselines.** We evaluate our MicroSeg and MicroSeg-M, with the comparisons with some classical methods in CIL (*e.g.*, Lwf-MC [19] and ILT [26]), as well as the state-of-the-art methods in CISS, including MiB [4], SDR [27], PLOP [11], and SSUL [5]. For the methods of CIL, we applied them to each experimental setup of CISS. We use the commonly used mean Intersection-over-Union (mIoU) as the evaluation metric for all the experimental evaluation and comparisons. In our experiment, the result of each experiment setting shows in three columns, *i.e.*, mIoU of classes learned in learning step $t = 1$ (*base classes*), in $t \geq 2$ (*novel classes*), and all classes to now, respectively. Besides comparing with prior methods, we also provides the experimental results of *joint* training, *i.e.*, training all classes offline. This setting is usually regarded as an upper bound of corresponding incremental scenario [4, 19].

## 4.2 Experimental Results

**Experiments on Pascal VOC 2012.** Besides the widely discussed scenarios of 19-1, 15-1, and 15-5 in previous works, we verified the performance of MicroSeg in several more challenging incremental scenarios, *e.g.*, scenarios of 5-3 (6 steps), 10-1 (11 steps), 2-2 (10 steps). These long-term incremental scenarios is more meaningful, and closer to situations in reality. Tab. 1 shows the final results after all learning steps in each incremental scenario. From the experiments, we can see that our approach has a very clear advantage over previous methods for all incremental scenarios. Even in the without-memory setting, MicroSeg shows performances over the state-of-the-art methods in almost all incremental scenarios. MicroSeg is obviously superior to SSUL with memory-free by the mIoU improvements of 2% to 3%, in the short-term incremental scenarios, *e.g.*, 19-1 and 15-5. While for the long-term scenarios, MicroSeg outperforms SSUL by a wider margin of up to over 7% improvement on mIoU. Not only that, MicroSeg-M (*i.e.*, MicroSeg with memory sampling of past samples) further strengthens the performances, and undoubtedly outperforms all prior methods in all scenarios.

Meanwhile, Fig. 3 (a) shows the change of average mIoU for all past classes under the scenarios of 15-1, with the training step progressing. In the first step, the performances of all methods are similar, but the mIoU starts to decrease as the learning steps increase because of the forgetting of the past classes. However, the mIoU of the previous work drops significantly, while MicroSeg effectively slows down the drop. By ensuring similar semantic segmentation capabilities to previous work, it demonstrates that MicroSeg exactly benefits incremental learning and outperforms prior works.

Table 2: Comparison with state-of-the-art methods on ADE20K.

| Method | ADE 100-5 (11 steps) | | | ADE 100-10 (6 steps) | | | ADE 100-50 (2 steps) | | | ADE 50-50 (3 steps) | | |
| --- | --- | --- | --- | --- | --- | --- | --- | --- | --- | --- | --- | --- |
| | 0-100 | 101-150 | all | 0-100 | 101-150 | all | 0-100 | 101-150 | all | 0-50 | 51-150 | all |
| ILT [26] | 0.1 | 1.3 | 0.5 | 0.1 | 3.1 | 1.1 | 18.3 | 14.4 | 17.0 | 3.5 | 12.9 | 9.7 |
| MiB [4] | 36.0 | 5.7 | 26.0 | 38.2 | 11.1 | 29.2 | 40.5 | 17.2 | 32.8 | 45.6 | 21.0 | 29.3 |
| PLOP [11] | 39.1 | 7.8 | 28.8 | 40.5 | 13.6 | 31.6 | 41.9 | 14.9 | 32.9 | 48.8 | 21.0 | 30.4 |
| SSUL[5] | 39.9 | 17.4 | 32.5 | 40.2 | 18.8 | 33.1 | 41.3 | 18.0 | 33.6 | 48.4 | 20.2 | 29.6 |
| SSUL-M[5] | 42.9 | 17.8 | 34.6 | 42.9 | 17.7 | 34.5 | 42.8 | 17.5 | 34.4 | 49.1 | 20.1 | 29.8 |
| MicroSeg (Ours) | 40.4 | 20.5 | 33.8 | 41.5 | 21.6 | 34.9 | 40.2 | 18.8 | 33.1 | 48.6 | **24.8** | **32.9** |
| MicroSeg-M (Ours) | **43.6** | **22.4** | **36.6** | **43.7** | **22.2** | **36.6** | **43.4** | **20.9** | **35.9** | 49.8 | 22.0 | 31.4 |
| Joint | 43.8 | 28.9 | 38.9 | 43.8 | 28.9 | 38.9 | 43.8 | 28.9 | 38.9 | 50.7 | 32.8 | 38.9 |

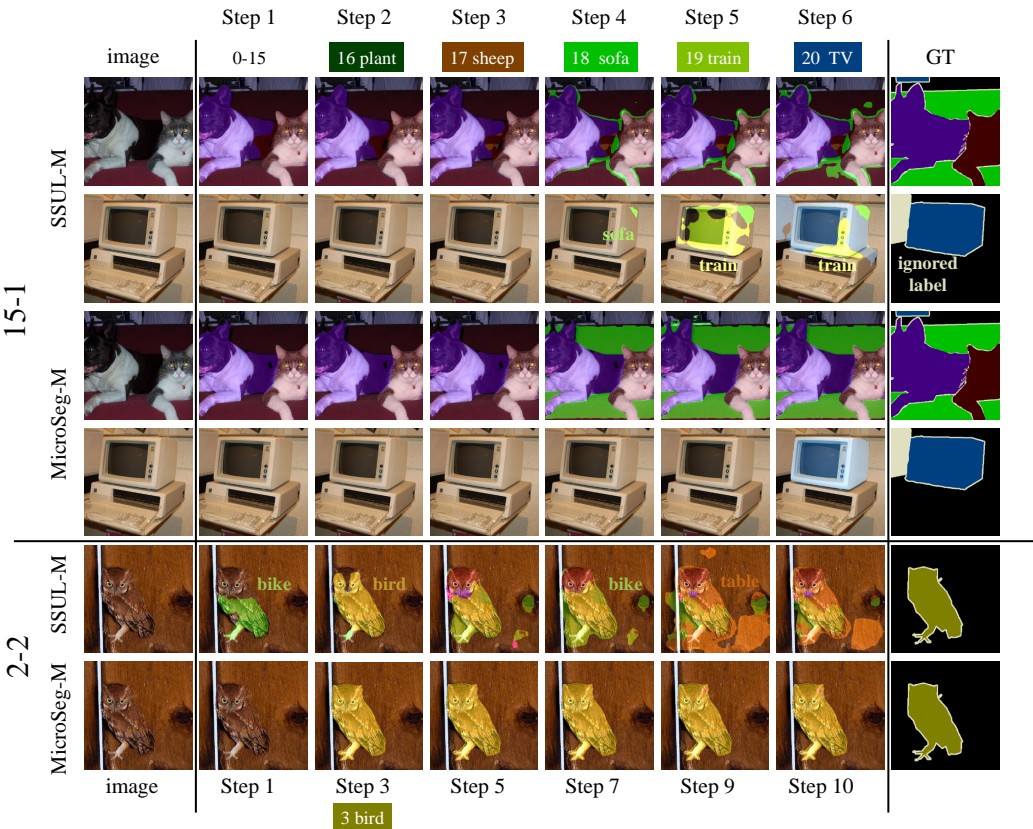

Figure 4: Qualitative analysis of MicroSeg on Pascal VOC 2012. Color-shaded boxes with semantic category and its index indicate the learned class during the corresponding step.

Fig. 3 (b) shows the similar analysis on the long-term scenario of 2-2. The effect of MicroSeg has shown a more significant advantage in such more challenging settings. Fig. 3 (c) shows the performance of MicroSeg with 20 random learning orders of 15-1 incremental scenario. Compared with prior approaches, our approach performs a better result with lower variance. The result illustrates that the robustness of our approach with various task settings outperforms prior methods.

**Experiments on ADE20K.** On the ADE20K dataset, we have provided experimental results with multiple incremental scenarios, which have various numbers of learning steps, and we extensively compared MicroSeg with prior methods in these setting. In Tab. 2, as in the case of Pascal VOC 2012, our MicroSeg has achieved a compelling enough performance without any samples sampling memory cost. Moreover, our MicroSeg-M extends this advantage and outperforms the previous state-of-the-art on all incremental scenarios. Further, the performance of our approach is very close to the offline training on base classes, and has a significant improvement to prior works on novel classes.

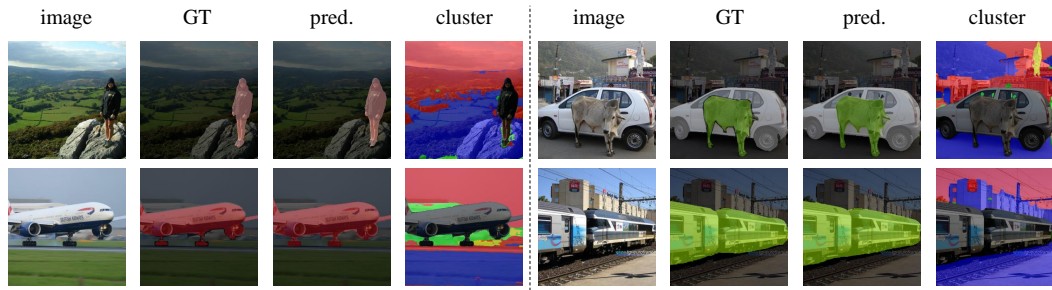

| image | GT | pred. | cluster | image | GT | pred. | cluster |

Figure 5: Qualitative performance of the proposed Micro mechanism. The first and second columns are sampled input images and ground truth. The third column shows the predictions of the last learning step. For the last column, distinct colors denotes different clusters of *unseen class*, *i.e.*, $c_{\hat{u},k}$ as mentioned in Sec. 3.3.

**Qualitative analysis of MicroSeg.** Fig. 4 shows qualitative analysis of our approach compared with SSUL-M, the current state-of-the-art method. Results are visualized of each learning step on multiple incremental scenarios of Pascal VOC 2012. First, we evaluated the typical scenario of 15-1 with a large number of base classes, to assess the plasticity of the model, *i.e.*, the ability to learn new classes. Regarding the samples set of {*dog*, *cat*, *sofa*} (the first and third rows), {*dog*, *cat*} belongs to the base classes while {*sofa*} is learnt in step 4. In contrast to SSUL-M, MicroSeg-M extract the feature of {*sofa*} well, and performs a complete segmentation of the novel class *sofa*. The other samples set (the second and fourth rows) is an image of *TV* (learned in the last learning step). The visualization indicated that, starting from step 4, SSUL-M tends to segment the images incorrectly into current foreground classes, *i.e.*, *sofa* and *train*, which are similar only in low-level semantic features (e.g., similar square shapes). Our approach, on the other hand, better learns the semantic features of the novel class by regional objectness, avoids the wrong segmentation, and segments it correctly and completely after learning the class *TV*.

We further evaluate the model stability (*i.e.*, ability to maintain old knowledge while learning new concepts) on the more challenging long-term scenario of 2-2. First SSUL-M incorrectly classifies *bird* in the image as *bike* at step 1 before learning *bird*. After learning class *bird*, SSUL-M products an acceptable result in step 3. However, in the subsequent learning step, the prediction results become more and more chaotic. Meanwhile, our MicroSeg has stronger stability, even in a long-term incremental scenario. MicroSeg maintains consistent results in multiple steps, and mitigates the negative effect of noise on the prediction with Micro mechanism.

**Qualitative analysis of the Micro mechanism** Fig. 5 shows the qualitative results of the proposed Micro mechanism. For the 'cluster' column, distinct colors denotes different clusters of *unseen class*, *i.e.*, $c_{\hat{u},k}$ motioned in Sec. 3.3. It can be seen from the figure, the background of samples are clustered well and reasonably with the Micro mechanism, even without supervision by ground truth. The mining of the unseen class not only mitigates the impact of the historical knowledge issue, but also benefits learning new concepts.

### 4.3 Ablation Study

**Effect of components in MicroSeg.**
Tab. 3 shows the contributions of two components of our approach, including proposal classification branch and Micro mechanism. The first row refers to the baseline and the last row stands for MicroSeg. The ablation studies are done on VOC 15-1. We will analyze them separately. The proposal classification branch improves the mIoU in the base class and the novel class, thus leading to an overall performance improvement.

Table 3: Ablation study of components in the proposed method on VOC 15-1, with DP (dense prediction branch), Proposal (proposal classification branch), and Micro (Micro mechanism). Numbers in brackets are the improvements over baseline (the first row).

| DP | Proposal | Micro | VOC 15-1 (6 tasks) | | |
|----|----------|-------|------|-------|------|
| | | | 0-15 | 16-20 | all |
| ✓ | ✗ | ✗ | 69.1 | 14.0 | 56.0 |
| ✓ | ✗ | ✓ | 76.8 | 31.2 | 65.9 (+9.9) |
| ✓ | ✓ | ✗ | 77.8 | 32.2 | 66.9 (+10.9) |
| ✓ | ✓ | ✓ | **80.1** | **36.8** | **69.8 (+13.8)** |

The usage of proposal classification branch increases both stability and plasticity, which are the most critical elements in incremental learning. As for Micro mechanism, observe the first and second rows of the table, the application of Micro mechanism decreases the influence of background shift, significantly improves the result. Due to the design of the Micro mechanism, the model generates more reasonable augmented labels and decreases the influence of background shift, strengthening the ability for acquiring novel classes and maintaining historical concepts. Finally, by combining the two components, MicroSeg can get better performance, 13.8% over the baseline.

**Effect of dense prediction branch.** Tab. 4 shows the effect of dense prediction branch. As we discussed in Sec. 3.4, this branch is only used when learning step $t = 1$. Hence, the constraint of dense prediction is only applied when the model is trained on the base classes. The result shows the comparison between the experiment of applying proposal classification branch only and using both branches. It can be observed that at the first learning step, dense prediction

Table 4: Ablation study of the dense prediction branch on VOC 15-1. DP: dense prediction branch, Proposal: proposal classification branch.

| Setting | step=1 | step=6 | | |
|---|---|---|---|---|
| | base | 0-15 | 16-20 | all |
| Proposal | 82.2 | 77.6 | 28.0 | 65.8 |
| Proposal+DP | **83.3** | **80.1** | **36.8** | **69.8** |

branch benefits the training on base classes. When all training steps are done, mIoU of novel classes is significantly improved. Dense prediction branch enables the model to be trained better on the base classes. Since MicroSeg applies remodeled labels composed of pseudo label, the quality of predictions of past classes becomes vital to the results. Thus, it increases the performance of the novel classes by improving the reliability of the remodeled label.

**Ablation of hyper-parameters.** Tab. 5 illustrates the influence of hyper-parameters: the number of future classes $K$ and threshold $\tau$ in label-remodeling. The results show that in most cases, our approach is not as sensitive to hyper-parameters.

Table 5: Search of hyper-parameters: the number of future classes $K$, and threshold $\tau$ in label-remodeling. All experiment are conducted on VOC 15-1.

| $K$ | 0-15 | 16-20 | all | $\tau$ | 0-15 | 16-20 | all |
|---|---|---|---|---|---|---|---|
| 1 | 78.6 | 30.9 | 67.2 | 0.1 | 79.7 | 30.7 | 68.3 |
| 3 | 79.7 | 34.0 | 68.8 | 0.3 | 81.1 | 31.3 | 69.2 |
| 5 | 80.1 | 36.8 | **69.8** | 0.5 | 80.8 | 33.1 | 69.4 |
| 7 | 80.4 | 32.8 | 69.1 | 0.7 | 80.1 | 36.8 | **69.8** |
| 9 | 80.1 | 33.1 | 68.9 | 0.9 | 80.0 | 32.5 | 68.7 |

## 5   Conclusions

In this study, we propose an effective method, MicroSeg, aiming at class incremental semantic segmentation problems by mining unseen classes via regional objectness. We first introduce proposals into class incremental semantic segmentation, and design a proposal-guided segmentation branch. To tackle background shift, we propose Micro mechanism, clustering and remodeling unseen classes, to capture diverse category features. Experiments demonstrate the effectiveness of our method. MicroSeg and MicroSeg-M achieve remarkable performance, especially on long-term incremental scenarios, outperforming prior state-of-the-art class incremental semantic segmentation methods.

**Limitations and societal impact.** Although our proposed Micro mechanism effectively mitigates background shift, catastrophic forgetting still exists as the learning step increases. As a simple baseline, we hope our work to inspire following-up research towards more challenging scenarios in incremental learning, *e.g.*, few base classes and long-term settings.

**Acknowledgment.** This work was supported in part by the National Key R&D Program of China (No.2021ZD0112100), the National Natural Science Foundation of China (No. 61972036), the Fundamental Research Funds for the Central Universities (No. K22RC00010).

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
