# Supplementary Materials for
# Mining Unseen Classes via Regional Objectness:
# A Simple Baseline for Incremental Segmentation

**Zekang Zhang**[1][*], **Guangyu Gao**[1][†], **Zhiyuan Fang**[1], **Jianbo Jiao**[2], **Yunchao Wei**[3,4]

[1] School of Computer Science, Beijing Institute of Technology
[2] School of Computer Science, University of Birmingham
[3] WEI Lab, Institute of Information Science, Beijing Jiaotong University
[4] Beijing Key Laboratory of Advanced Information Science and Network
zkzhang1998@outlook.com

## 1    Details for Reproducibility

**Protocols.**    Here we present more details of the *overlapped* and *disjoint* settings. Specifically, the *overlapped* setting means pixels in samples from $\mathcal{D}_t$ can belong to any classes, including past (step 1 to $t-1$, formally, $\mathcal{C}_{1:t-1}$), now ($\mathcal{C}_t$) and future ($\mathcal{C}_{t+1:T}$). Note that only the classes in $\mathcal{C}_t$ would be annotated in $\mathcal{Y}_t$. Besides, images with multiple classes would appear in several learning steps, with different annotations. The *disjoint* setting, studied by [1, 11], has a non-overlapped $\{\mathcal{D}_t\}$. Thus, each learning step contains a unique $\mathcal{D}_t$, whose pixels only belong to classes seen in $\mathcal{C}_{1:t-1}$ or in $\mathcal{C}_t$. It is obvious that *overlapped* setup is more realistic, because *overlapped* has a weaker restriction on data than *disjoint*.

**Details of Proposal Reorganization**    The logits $\boldsymbol{p} \in \mathbb{R}^{N \times |\mathcal{C}|}$ is generated by classification. While proposal $\boldsymbol{P} \in \{0,1\}^{N \times H \times W}$ is a group of binary masks where $N$ is the number of masks and $H, W$ denote height and width. Proposal reorganization is a matrix multiplication of logits and proposal, to obtain probability map of image, with the shape of $|\mathcal{C}| \times H \times W$.

**Details of MicroSeg-M.**    As mentioned in the main paper, MicroSeg-M is an advanced version of MicroSeg by introducing memory sampling strategy from [2]. To be more specific, the strategy is based on random-sampling, and advanced ensuring that there is at least one sample of every seen class in the memory.

## 2    More Experimental Results and Discussions

**Detailed experimental results.**    Tab. 1 provides experimental results of Pascal VOC 2012 [5] of different incremental scenarios with each class.

**Experimental results of disjoint setup.**    As explained in Sec. 1, disjoint setup is not realistic due to its non-overlapping samples in each learning step. However, we still provide experimental results of the disjoint setup, for a fair comparison with prior works [1, 2, 4, 11]. The result of Tab. 2 shows that our proposed methods, *i.e.*, MicroSeg and MicroSeg-M, outperform the state-of-the-art methods.

---

[*]Work done during an internship at WEI Lab of Beijing Jiaotong University.
[†]Corresponding author.

36th Conference on Neural Information Processing Systems (NeurIPS 2022).

Table 1: Detailed experimental results of Pascal VOC 2012 with class.

| | bg | aero | bike | bird | boat | bottle | bus | car | cat | chair | cow | table | dog | horse | mbike | person | plant | sheep | sofa | train | TV | mIoU |
|---|---|---|---|---|---|---|---|---|---|---|---|---|---|---|---|---|---|---|---|---|---|---|
| **10-1 (11 steps)** | | | | | | | | | | | | | | | | | | | | | | |
| MicroSeg | 89.7 | 91.6 | 41.2 | 95.1 | 74.1 | 84.4 | 90.3 | 89.6 | 86.8 | 37.4 | 18.6 | 33.4 | 75.1 | 38.5 | 76.6 | 87.9 | 15.7 | 34.1 | 28.5 | 60.0 | 37.0 | 61.2 |
| MicroSeg-M | 90.9 | 86.8 | 42.0 | 91.5 | 78.0 | 84.0 | 90.9 | 87.6 | 89.3 | 40.1 | 68.8 | 47.5 | 69.4 | 40.4 | 84.9 | 85.7 | 34.0 | 45.2 | 33.6 | 70.8 | 61.6 | 67.8 |
| **2-2 (10 steps)** | | | | | | | | | | | | | | | | | | | | | | |
| MicroSeg | 85.7 | 73.9 | 24.4 | 40.8 | 25.9 | 48.8 | 27.7 | 74.9 | 48.0 | 4.7 | 39.3 | 5.1 | 54.8 | 44.5 | 67.1 | 80.5 | 11.8 | 38.0 | 27.0 | 45.2 | 47.0 | 43.5 |
| MicroSeg-M | 87.2 | 72.8 | 20.2 | 39.2 | 34.2 | 59.1 | 79.5 | 82.1 | 78.3 | 6.6 | 35.3 | 13.5 | 69.4 | 47.9 | 69.7 | 80.1 | 25.9 | 56.0 | 28.1 | 70.5 | 41.0 | 52.2 |
| **15-1 (6 steps)** | | | | | | | | | | | | | | | | | | | | | | |
| MicroSeg | 89.8 | 92.9 | 44.4 | 95.5 | 76.0 | 82.8 | 92.9 | 90.9 | 93.4 | 48.6 | 68.2 | 52.9 | 89.8 | 83.5 | 89.7 | 90.4 | 11.7 | 56.2 | 24.8 | 57.3 | 34.2 | 69.8 |
| MicroSeg-M | 91.2 | 93.2 | 43.4 | 93.1 | 78.7 | 84.7 | 91.9 | 91.0 | 94.3 | 45.8 | 88.4 | 56.0 | 88.5 | 84.6 | 86.9 | 90.1 | 22.9 | 71.5 | 31.0 | 71.5 | 65.6 | 74.4 |
| **5-3 (6 steps)** | | | | | | | | | | | | | | | | | | | | | | |
| MicroSeg | 89.1 | 91.2 | 34.8 | 89.3 | 73.9 | 87.3 | 82.3 | 83.6 | 79.6 | 4.6 | 55.5 | 40.0 | 76.6 | 55.5 | 74.1 | 87.9 | 31.2 | 51.4 | 30.5 | 70.2 | 62.7 | 64.3 |
| MicroSeg-M | 90.9 | 81.2 | 34.7 | 87.8 | 67.4 | 87.0 | 85.0 | 78.2 | 80.6 | 21.8 | 55.1 | 37.9 | 77.3 | 52.9 | 74.5 | 86.2 | 43.4 | 38.8 | 33.8 | 74.8 | 67.0 | 64.6 |
| **19-1 (2 steps)** | | | | | | | | | | | | | | | | | | | | | | |
| MicroSeg | 88.8 | 87.5 | 41.4 | 92.5 | 72.6 | 85.3 | 96.0 | 85.6 | 94.9 | 36.6 | 92.4 | 57.6 | 90.9 | 89.6 | 85.0 | 89.9 | 71.6 | 82.2 | 50.4 | 87.0 | 13.9 | 75.7 |
| MicroSeg-M | 93.6 | 86.8 | 43.7 | 87.5 | 73.3 | 84.0 | 96.1 | 90.5 | 93.8 | 37.3 | 87.8 | 62.9 | 90.4 | 88.6 | 88.6 | 89.9 | 71.0 | 83.0 | 51.5 | 85.9 | 62.9 | 78.5 |
| **15-5 (2 steps)** | | | | | | | | | | | | | | | | | | | | | | |
| MicroSeg | 91.7 | 91.1 | 44.8 | 92.4 | 80.1 | 85.2 | 93.3 | 91.3 | 94.5 | 43.0 | 69.3 | 55.9 | 90.5 | 83.9 | 90.1 | 90.1 | 32.3 | 62.3 | 30.6 | 82.0 | 57.0 | 73.8 |
| MicroSeg-M | 93.0 | 91.5 | 42.9 | 94.0 | 80.1 | 85.4 | 92.2 | 91.8 | 94.7 | 44.5 | 89.5 | 55.2 | 90.1 | 87.8 | 90.3 | 90.5 | 48.0 | 76.5 | 36.1 | 77.6 | 58.0 | 76.6 |

Table 2: Experimental results on Pascal VOC 2012 for *disjoint* setup.

| Method | VOC 15-1 (6 steps) | | | VOC 19-1 (2 steps) | | | VOC 15-5 (2 steps) | | |
|---|---|---|---|---|---|---|---|---|---|
| | 0-15 | 16-20 | all | 0-19 | 20 | all | 0-15 | 16-20 | all |
| LwF-MC [9] | 4.5 | 7.0 | 5.2 | 63.0 | 13.2 | 60.5 | 67.2 | 41.2 | 60.7 |
| ILT [10] | 3.7 | 5.7 | 4.2 | 69.1 | 16.4 | 66.4 | 63.2 | 39.5 | 57.3 |
| MiB [1] | 46.2 | 12.9 | 37.9 | 69.6 | 25.6 | 67.4 | 71.8 | 43.3 | 64.7 |
| SDR [11] | 59.4 | 14.3 | 48.7 | 70.8 | 31.4 | 68.9 | 74.6 | 44.1 | 67.3 |
| PLOP [4] | 57.9 | 13.7 | 46.5 | 75.4 | 38.9 | 73.6 | 71.0 | 42.8 | 64.3 |
| SSUL [2] | 74.0 | 32.2 | 64.0 | 77.4 | 22.4 | 74.8 | 76.4 | 45.6 | 69.1 |
| SSUL-M [2] | 76.5 | 43.4 | 68.6 | 77.6 | 43.9 | 76.0 | 76.5 | 48.6 | 69.8 |
| MicroSeg (Ours) | 73.7 | 24.1 | 61.9 | 80.6 | 16.0 | 77.4 | 77.4 | 43.4 | 69.3 |
| MicroSeg-M (Ours) | **80.0** | **47.6** | **72.3** | **81.1** | **45.1** | **79.4** | **80.7** | **55.2** | **74.7** |
| Joint | 82.7 | 75.0 | 80.9 | 81.0 | 79.1 | 80.9 | 82.7 | 75.0 | 80.9 |

**Further discussion of the proposal.** To further investigate the effect of the proposal, we conducted experiments with modified SSUL [2] (a prior work of CISS). Specifically, SSUL performs the mining of the background by using saliency map [8] as a *saliency detector*, to extract unknown class from background. We complete the experiments on replacing the saliency map with segment proposal and ground truth. The qualitative analysis of these three kinds of "saliency detectors" is shown in Fig. 1. What can be observed is, saliency map only focuses on the most significant areas of the image (*people*), while some areas are missed (*bottle*), proposal tends to focus on more area (*plants*), as the area circled in the Fig. 1. The ground truth (GT), on the other hand, figures out exactly the area that needs to be segmented. Therefore, GT can be considered as the upper bound of quality of the saliency map, because it has complete and precise foreground annotation.

We provide experimental results of VOC 15-1 and VOC 19-1 scenarios in Tab. 3. From the results we can observe that, compared with saliency map, GT can significantly improve the performance of SSUL, while proposal can only slightly improve it. As a saliency detector, *proposal* does not work well compared to *saliency map*. It indicates that the performance improvement of our method comes from neither the information contained in proposal, nor its positive impact on the performance of semantic segmentation only. The key of our proposed method is proposals with the Micro mechanism, trying to solve the problem of background shift in incremental learning.

On the other hand, we further discussed B from the perspective of methods of generating proposals. We have conducted the experiments of replacing proposal generator, including MaskFormer [3] and RPN in Mask R-CNN combined with class-agnostic segmentation head [6, 7] (denote as RPN+Seghead). We also conduct the results for generating different numbers of proposals ($N$) with Mask2Former. Note that the original setting of MicroSeg is Mask2Former ($N = 100$). All experiments are done in VOC 15-1. As shown in Tab. 4, using other methods as a proposal generator does not significantly affect the performance, comparing to the original setting of MicroSeg. This result further supports our previous claims about propsoals.

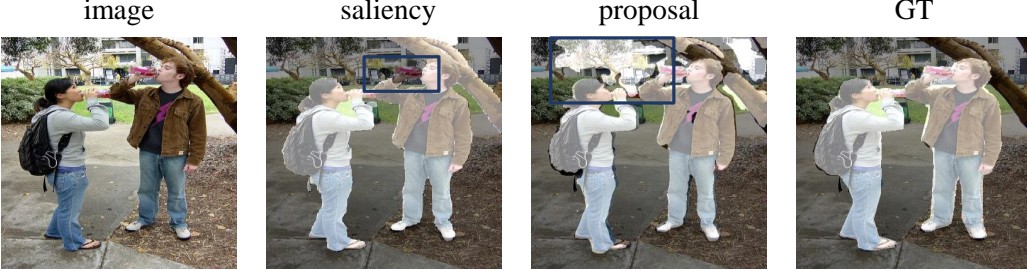

| image | saliency | proposal | GT |

Figure 1: Qualitative comparison of different saliency detector. The boxes point out the main differences among them.

Table 3: Experimental results of SSUL, using different saliency detector. Each row denotes a saliency detector type that SSUL applies: saliency map, saliency proposal and ground truth. The row *saliency map* represents the results of original settings of SSUL. And the following two rows shows the results of SSUL with modified saliency detector. We provide the experimental results of VOC 15-1 and VOC 19-1.

| setting | VOC 15-1 (6 steps) | | | VOC 19-1 (2 steps) | | |
|---|---|---|---|---|---|---|
| | 0-15 | 16-20 | all | 0-19 | 20 | all |
| saliency | 77.3 | 36.6 | 67.6 | 77.7 | 29.7 | 75.4 |
| proposal | 76.5 | 38.8 | 67.5 | 78.6 | 35.0 | 76.6 |
| GT | 76.9 | 46.0 | 69.7 | 79.4 | 65.1 | 78.8 |

Table 4: Experimental results of MicroSeg with different proposal generator and different numbers of proposals. We provide the experimental results of VOC 15-1.

| setting | VOC 15-1 (6 steps) | | |
|---|---|---|---|
| | 0-15 | 16-20 | all |
| RPN+Seghead | 80.5 | 33.0 | 69.2 |
| MaskFormer($N = 100$) | 79.3 | 34.1 | 68.5 |
| Mask2Former($N = 100$)(Ours) | 80.1 | 36.8 | 69.8 |
| Mask2Former($N = 200$) | 80.4 | 37.3 | 70.1 |

**Further discussion of the selection of $K$.** Here we will illustrate the reason for $K = 1$ in ADE20K dataset. Hyper-paramenter $K$ is the number of future classes. When the prediction of the historical classes is accurate (i.e., high accuracy like 82.7 for VOC 15-1), the remaining uncertainty regions are mainly future classes, and as a result a larger $K$ can better distinguish different future classes. But if the prediction of historical classes is not that accurate (e.g., 43.8 as for the ADE 100-10), many historical classes would be mis-classified and grouped into the remaining future class regions. In this case, if the $K$ is still set to a large value, pixels belonging to the same historical class will be

Table 5: Discussion of $K$ and feature extractor. Using a stronger backbone, a larger $K$ in micro mechanism achieved positive results.

| backbone | $K$ | ADE 100-10 (6 steps) | | |
|---|---|---|---|---|
| | | 0-15 | 16-20 | all |
| ResNet-101 | 1 | 41.5 | 21.6 | 34.9 |
| | 2 | 41.1 | 12.4 | 31.6 (-3.3) |
| Swin Transformer | 1 | 43.4 | 29.6 | 38.8 |
| | 2 | 43.7 | 29.4 | 39.0 (+0.2) |

forced to be separated into different categories and as a result confuse the model learning. In this case (*e.g.*on the ADE20K dataset), choice of a smaller $K$ (*i.e.*, $K = 1$) would be safer and less harmful.

Meanwhile, on the ADE20K dataset, in order to conduct a fair comparison with prior CISS methods [1, 2, 4], we choose to use the ResNet-101 as the backbone. While if we adapt a more advanced model (e.g. the Swin Transformer), a larger $K$ in micro mechanism achieves positive results. The experimental results are provided in Tab. 5.

## 3 More Qualitative Results

### 3.1 Qualitative analysis of ADE20K

Fig. 2 shows the qualitative results of the proposed MicroSeg on ADE20K dataset [12] with the 100-10 scenario. Overall, our method performs well in CISS on ADE20K dataset. The first two rows show the ability of the MicroSeg of learning new concepts (*dishwasher* is learned at step 4 and *fan* is learned at step 5). The following two rows show the stability of our method. With the learning step increasing, MicroSeg performs stable predictions. Historical knowledge is not forgotten when learning new concepts. Moreover, for ADE20K, a dense-labeled dataset, both thing and stuff classes are annotated. Our method still achieves well performance at segmenting samples with only stuff classes, *e.g.*, *sky*, *grassland*, *lake*, as shown in the last two rows.

### 3.2 Extra qualitative results of MicroSeg

We provide extra qualitative results of ADE20K 100-10 and VOC 15-1 in Fig. 3 and Fig. 4.

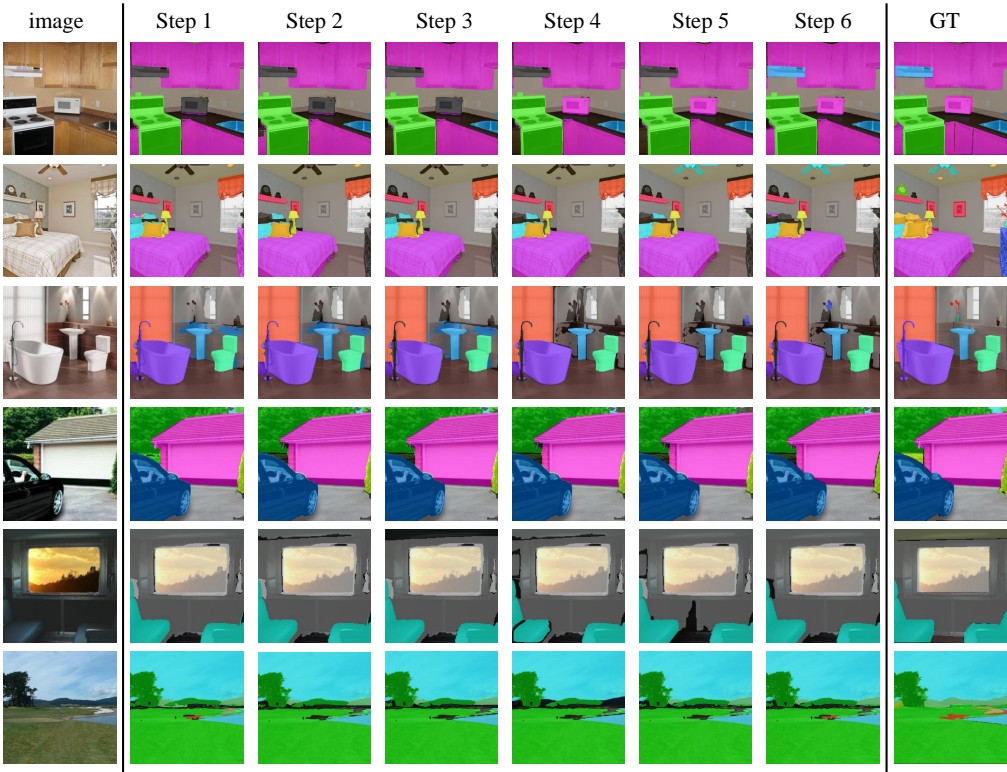

Figure 2: Qualitative results on the ADE20K dataset with the 100-10 scenario.

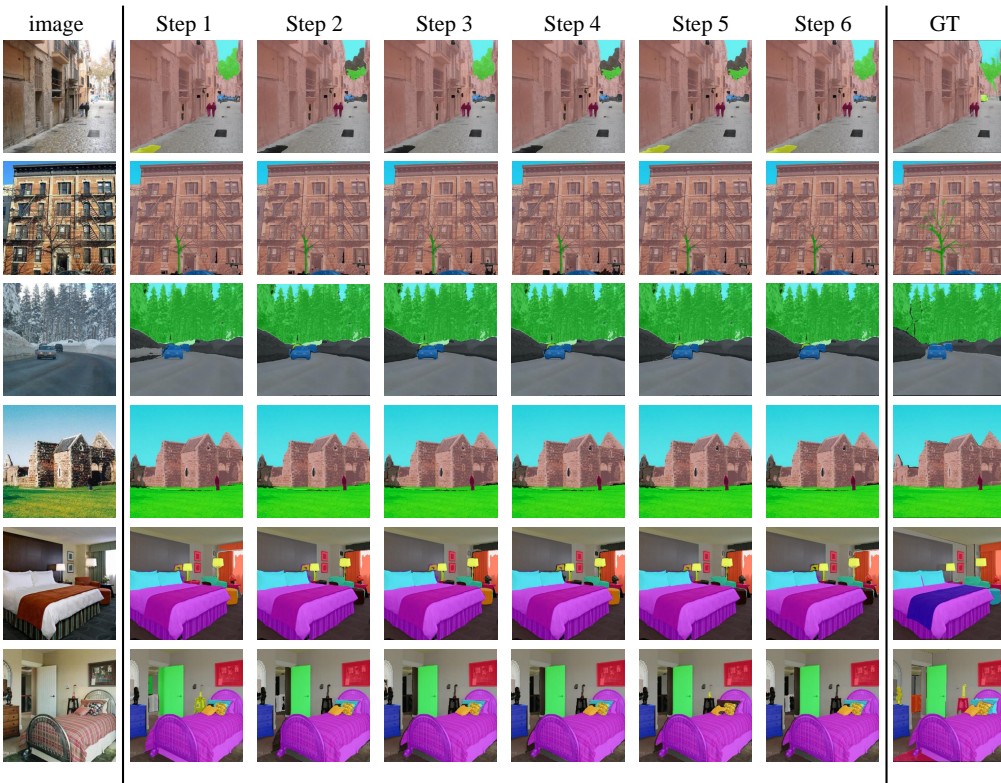

Figure 3: Extra qualitative results on ADE20K dataset with the 100-10 scenario.

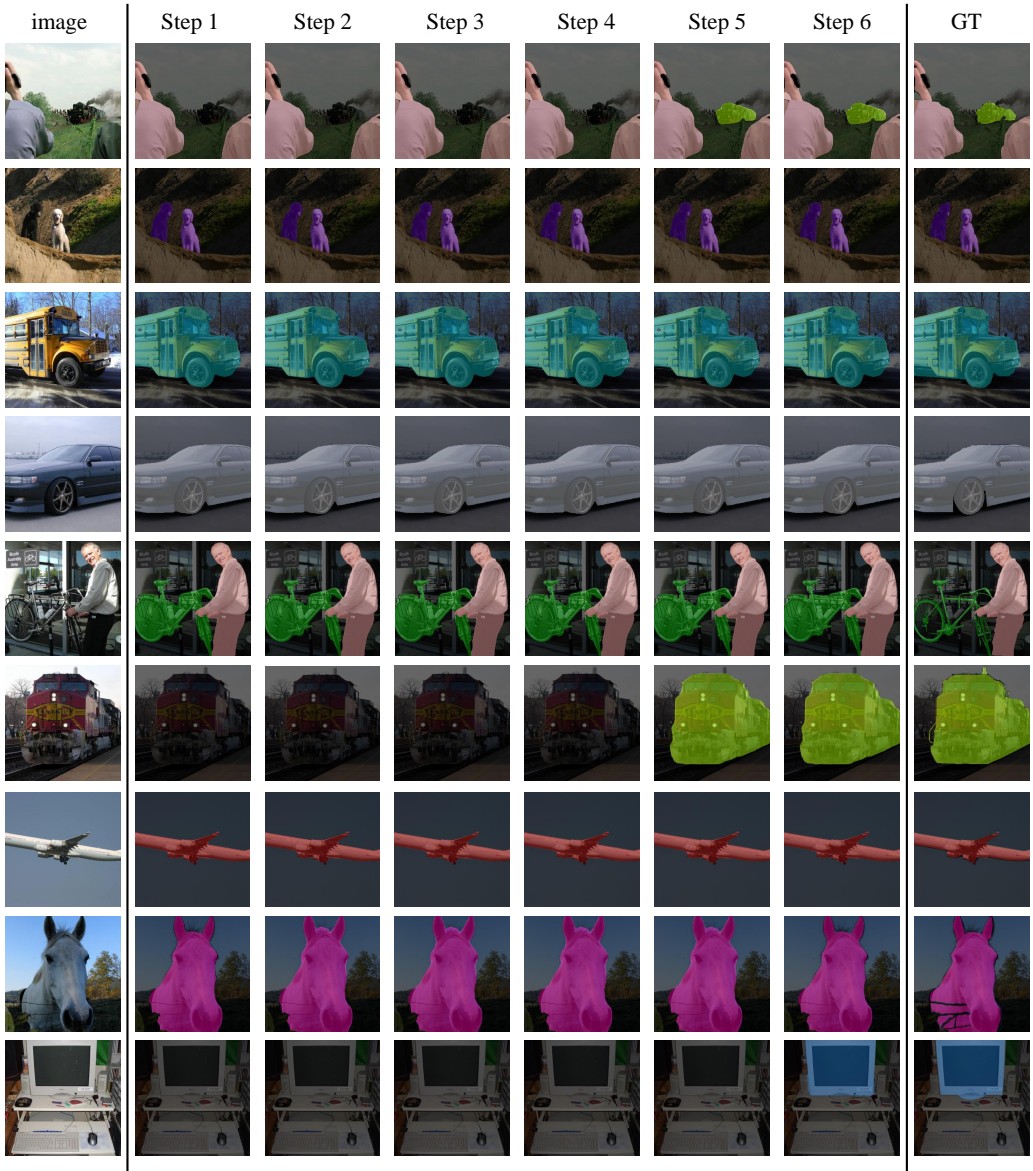

Figure 4: Extra qualitative results on Pascal VOC 2012 dataset with the 15-1 scenario.