# OpenReview forum: "Mining Unseen Classes via Regional Objectness: A Simple Baseline for Incremental Segmentation"
_NeurIPS.cc/2022/Conference — NeurIPS 2022 Accept_

### Official Review · Reviewer_3V6U · 2022-06-20

**Rating:** 4
**Confidence:** 4
**Soundness:** 3 good
**Presentation:** 2 fair
**Contribution:** 2 fair

**Summary:**

This paper proposes a new method for class-incremental segmentation. Based on the assumption that background regions with strong objectness possibly belong to those concepts in the historical or future stages, this paper introduces a proposal generator for generating multiple segment proposals. Also, some tricks including label remodeling and the new loss functions are designed. The proposed method gains good performance on Pascal VOC and ADE20K.

**Questions:**

Please refer to the weaknesses part for details.

I cannot recommend accepting this work at its current version considering the high standards of NeurIPS. However, considering the interesting motivation and the relatively novel method, I suggest the authors to carefully revise the paper. I believe it would be a good work if the writing is clearer.

**Strengths And Weaknesses:**

I appreciate the motivation of this work, which is well presented in the introduction. Also, I believe the proposed method makes sense and is somehow novel.

My main concern of this work is the writing. I suggest the authors to reconstruct this paper in a clearer form. Many parts of this paper are quite confusing and hard to understand. Also, it lacks a lot of details, so that it is hard for others to follow. Below are some of the weaknesses:

1. This paper proposes a proposal generator. However, the detailed structure of the generator is not clear.

2. In Line 142 to Line 143, 'Then $P$  is *reorganized* to the original shape and the prediction scores are assigned accordingly'. I can understand this after checking Figure 2. However, I think it may be better if the authors illustrate this step more clearly by words or equations.

3. Why the contrastive loss (Equation 7) is used for optimization? The motivation and prupose are not clear.

4. What's the value of $\tau$ in Equation 3? It is not illustrated in the implementation details.

5. Some ablation experiments are missed. For example, how the settings of hyper-parameters $N$, $K$ and $\tau$  affect the performance?

---

> ### Author Response · Authors · 2022-08-02
> **Response to Reviewer 3V6U**
>
> ### *Responses to weaknesses*
>
>
> 0. **[My main concern of this work is the writing. I believe it would be a good work if the writing is clearer]**
> Thanks for acknowledging the contributions of our paper and the valuable suggestions! We will carefully proofread and polish the paper to address the writing issues in the revised version.
>
> 1. **[More details about proposal generator]**
> The structure of the proposal generator is identical to Mask2Former. As we described in L210 to L212 in the main paper, we use the off-the-shelf Mask2Former pretrained on MS-COCO to generate proposals. It is not optimized during the training stage. MicroSeg aims to accomplish the CISS with regional objectness information rather than the mask proposal generator. We will consider explaining it clearly in a revised version.
> 3. **[Detailed descriptions of proposal reorganization]**
> Thanks for the suggestion, we will reformulate it by following words:
>
>
>     The logits $p \\in \\mathbb{R}^{N \\times | \\mathcal{C}_{1:t} | }$ is generated by classification.
>
>     While proposal is a group of binary masks ${P} \\in \\{0,1\\}^{N \\times H \\times W}$ where $N$ is the number of masks and $H, W$ denote height and width. Proposal reorganization is a matrix multiplication of logits and proposal,  to obtain probability map of image, with the shape of ${| \\mathcal{C}_{1:t}| \\times H \\times W}$. We will make the description of proposal reorganization clearer in the revised version.
>
>
>
>
> 4. **[Detailed descriptions of contrastive loss]**
> Due to the length limitation of the paper, we only briefly introduce the contrastive loss in Sec. 3.4 in main paper, and we will add more descriptions about the component:
> Contrastive loss is used to keep the effectiveness of Micro mechanism. In Micro mechanism, we represent future classes $c_{\hat{u}}$ as a set of K classes to better cache the feature diversity in those loose future classes. To avoid model degradation caused by $c_{\hat{u},k}$ characterizing the same features, we use unsupervised contrastive loss to force them to capture the diverse concepts, thus ensuring the effectiveness of the Micro mechanism. *Cluster* columns of Fig. 2, in Supplementary Material, show the qualitative performance of $c_{\hat{u},k}$ to capture diverse contexts.
> 6. **[The value of hyper-parameter $\tau$ in Equation 3]**
> $\tau$ is set to 0.7 in our experimental setting. We will add the hyper-parameter to implementation details of our paper in the revised version.
> 8. **[Extra ablation experiments]**
> We have conducted the ablation studies of $\mathit{K}$, $\mathit{N}$, $\tau$. All experiments are done in VOC 15-1.
>     The ablation study for the hyper-parameter $\mathit{N}$:
>
>     |    $\mathit{N}$                | base | novel |    all     | #params. |
>     |:------------------:|:----:|:-----:|:----------:|:--------:|
>     | 100 | 80.1 | 36.8  |    69.8    |   107M   |
>     | 200 | 80.4 | 37.3  | 70.1 |   216M   |
>
>     As the proposal generator model size grows, the performance of MicroSeg improves as well. We choose Mask2Former(N=100) as a better trade-off between model size and performance.
>
>     The ablation study for hyper-parameter $\mathit{K}$ in Micro mechanism:
>
>     | $\mathit{K}$ | base | novel | all  |
>     |:------------:|:----:|:-----:|:----:|
>     |      1       | 78.6 | 30.9  | 67.2 |
>     |      3       | 79.7 | 34.0  | 68.8 |
>     |      5       | 80.1 | 36.8  | **69.8** |
>     |      7       | 80.4 | 32.8  | 69.1 |
>
>     So we choose $\mathit{K}$ = 5 in VOC2012 for best performance.
>
>
>
>     The ablation study for hyper-parameter $\tau$:
>
>     | $\tau$ | base | novel | all  |
>     |:------:|:----:|:-----:|:----:|
>     |  0.1   | 79.7 | 30.7  | 68.3 |
>     |  0.3   | 81.1 | 31.3  | 69.2 |
>     |  0.5   | 80.8 | 33.1  | 69.4 |
>     |  0.7   | 80.1 | 36.8  | **69.8** |
>     |  0.9   | 80.0 | 32.5  | 68.7 |
>
>     We choose $\tau$ = 0.7 for best performance.

---

> > ### Comment · Reviewer_3V6U · 2022-08-07
> > **My Remaining Concerns**
> >
> > Thanks for the authors' clear response. I have read the response from authors and comments from other reviewers.
> >
> > My remaining concern is about the hyper-parameters. It seems the proposed method relies on a lot of hyper-parameters ($N$, $K$, $\tau$). Moreover, the method performance seems to be quite sensitive to $\tau$, ($\tau$ from 0.7 to 0.9, performance from 69.8 to 68.7 ).
> >
> > I also have a concern about the $K$ for ADE20K, where it is set as 1. The proposed micro mechanism seems to rely on assembling multiple    logits of different $c_{\hat{u},k}$. So can it still work when $K$ is 1? I notice a similar concern by Reviewer uMp6. However, I think the authors' response not clear enough. Is there any further explanation?

---

> > > ### Author Response · Authors · 2022-08-09
> > > **Response to Reviewer 3V6U**
> > >
> > > Thank you for your feedback! For your remaining concern:
> > >
> > >
> > > **[Concerns about hyper-parameters]**
> > >
> > > There may be a misunderstanding. $N$ is the number of the mask proposals, which is inherited from Mask2Former and fixed as 100 for all experiments. Therefore, only the hyper-parameters $K$ and $\tau$ will influence the performance of MicroSeg. For $K$, we agree that it is sensitive to the difficulty of the given dataset. However, we consider that an expected $K$ is easy to be found according to some experimental analysis (more details are given in the following response). The value of $\tau$ represents the threshold of pseudo-label, we have provided the results of different settings in the table below, and the fluctuation range of performance is from 68.3 to 69.8. This range is actually reasonable since the worst one (68.3 as $\tau$ = 0.1) still outperforms the previous SOTA (67.6, compared to SSUL).
> > >
> > > | $\tau$ | base | novel | all  |
> > > |:------:|:----:|:-----:|:----:|
> > > |  0.1   | 79.7 | 30.7  | 68.3 |
> > > |  0.3   | 81.1 | 31.3  | 69.2 |
> > > |  0.5   | 80.8 | 33.1  | 69.4 |
> > > |  0.7   | 80.1 | 36.8  | **69.8** |
> > > |  0.9   | 80.0 | 32.5  | 68.7 |
> > >
> > > **[Can Micro mechanism still work when K is 1? & Detailed explanation about the choice of $K$]**
> > >
> > > First, we would like to clarify the definition of $K$, which is the number of *future classes*. When the prediction of the historical classes is accurate (i.e., high accuracy like 82.7 for VOC 15-1), the remaining uncertainty regions are mainly future classes, and as a result a larger $K$ can better distinguish different future classes. But if the prediction of historical classes is not that accurate (e.g., 43.8 as for the ADE 100-10), many historical classes would be mis-classified and grouped into the remaining future class regions. In this case, if the $K$ is still set to a large value (>1), pixels belonging to the same historical class will be forced to be separated into different categories and as a result confuse the model learning. In this case, a smaller $K$ (i.e., $K = 1$) would be safer and less harmful. We agree that the Micro mechanism is indeed incomplete with $K = 1$: the unsupervised contrastive loss does not work for $K = 1$, but there is still supervised BCE loss to constrain $c_{\hat{u}}$ in the Micro mechanism. And it is worth noting that even in this case, the proposed MicroSeg still achieves SOTA performance (i.e., 36.3 v.s. 33.1, compared to SSUL, in ADE 100-10).
> > >
> > > On the ADE20K dataset, in order to conduct a fair comparison with prior CISS methods [4, 5, 11], we choose to use the ResNet101 and DeeplabV3. But we assume a more advanced model (e.g. the Swin Transformer) with a larger $K$ would lead to a different conclusion. Due to the time limit of the rebuttal, we will include this experiment in the revised version to better illustrate the choice of $K$.

---

> > > > ### Comment · Reviewer_3V6U · 2022-08-09
> > > > **Thanks for the response**
> > > >
> > > > Thanks for the detailed response. I think the response addresses part of my concerns. But my concern on $K$ still remains. $K$ denotes the the number of future classes. So intutively $K$ for ADE20K should be larger than that of VOC, since ADE20K usually has more classes in one image. Therefore, the setttings of $K$ for different datasets are quite counterintuitive, though I can understand that a smaller $K$ may be better if the prediction of historical classes is not accurate enough. I think this indicates a defect of the proposed micro mechanism.

---

> > > > > ### Author Response · Authors · 2022-08-10
> > > > > **Response to the remaining concern**
> > > > >
> > > > > Thanks for your further response and acknowledging our efforts. Intuitively, $K$ for ADE20K should be larger than that of VOC, since one image is often with more classes in ADE20K. However, it is worth noting that the number of future classes in ADE20K is of the same order of magnitude as in VOC in a single image based on the current setting, which is also shared in prior CISS works [4, 5, 11]. We show the comparison in the table below. In this case, the accuracy of historical classes will mainly dominate how to select $K$. We hope this observation can further help address your concern.
> > > > >
> > > > >
> > > > > | Dataset | Avg all classes | Avg future classes |
> > > > > |:-------:|:---------------:|:------------------:|
> > > > > | ADE20K  |      10.58      |        1.67        |
> > > > > |   VOC   |      2.76       |        1.04        |

---

### Official Review · Reviewer_uMp6 · 2022-06-29

**Rating:** 7
**Confidence:** 4
**Soundness:** 3 good
**Presentation:** 3 good
**Contribution:** 3 good

**Summary:**

The authors propose a novel way to cluster the unseen classes present in the background using K different background components and a pre-trained class-agnostic mask proposal generator (Mask2Former). The results show a clear improvement from pixel-wise methods, which is further improved by the micro mechanism.


**Questions:**

- The disjoint setting as it is described would allow future classes unlabeled in the background contrary to the definition of disjoint in MiB [4]. Is this an actual change of the setting or just a mistake in the description?
- Are the joint results on ADE20k based on the proposed architecture, including the Mask2Former or for the base DeepLabv3 model (the results are identical to those reported in [4], which seems low if the Mask2Former also is used since they achieve 47.7 using ResNet50 by training on ADE20k)?
- Please specify which pre-trained weights that are used for the ResNet-101 network for facilitating reproducibility
- How does the proposed split of segment proposals and classification differ from [8]? And is the splitting of the task into segment proposal and classification novel for semantic segmentation or for CISS?
- Does the performance of the proposal generator differ between classes present and absent in MS-COCO?
- Is the contrastive loss only applied to the background? And if so, is this the background before or after pseudo-labeling?
- Which losses are applied to the baseline in Tab 3? It seems to perform better than all methods except for SSUL by only using pseudo-labels?

**Limitations:**

Yes, though there would be room to further discuss the importance of the pre-training of the proposal generation and possible limitations of it when studying data which differs more than COCO/Pascal/ADE20k.

**Strengths And Weaknesses:**

Strengths:
- Clear improvement in the benchmarks over current SOTA
- Interesting approach for mining unseen classes and perform pseudo clustering of these
- Good comparison of how stable the performance is under different class orders on Pascal (Fig 3c).

Weaknesses:

Some claims of novelty seems exaggerated and require revision/clarification, i.e.
- Experimental settings:
  - The different settings that are "introduced" are not new, i.e.
    - Pascal VOC 10-1 and ADE20k 100-5 were used in PLOP [11]
    - Pascal VOC 10-1 + 5-3 and ADE20k 100-5 were all used in SSUL [5]
  - "RECALL: Replay-based Continual Learning in Semantic Segmentation" (ICCV2021) has not been referenced in the paper though they also proposed the 10-1 setting as well as 10-10 and 10-5.
- Mask2Former:
  - Partly unclear which parts of the proposal generation and classification that differs from Mask2Former for semantic segmentation.

No ablation study of how the number of components in the Micro mechanism affects the results, which would be interesting since this is the main contribution of the paper. The choice of K=1 for ADE20k seems conter-intuitive, since ADE20k contains much more classes than Pascal and clustering these into a single component would enforce similar feature representations for all unseen classes.

It would have been interesting to see how the performance varies for different class orders on ADE20k since the class-frequency decreases drastically between stuff and things and as the original index increases. (As done in MiB and SDR)

---

> ### Author Response · Authors · 2022-08-02
> **Response to Reviewer uMp6**
>
> ### *Responses to weaknesses*
> 1. **[Some claims of novelty require revision/clarification]**
> Thanks for pointing out this, we will revise the descriptions and add references accordingly. The structure of our proposal generator is identical to Mask2Former. It should be noted that we only use an off-the-shelf pretrained Mask2Former model as a proposal generator. It is not optimized during the training stage. And proposal classification is not related to Mask2Former, as shown in Fig. 2 in the main paper.
>
> 2. **[Ablation study of $\mathit{K}$ and choice of $\mathit{K}$ in ADE20K]**
> As suggested, here we provide the ablation study of $\mathit{K}$ in Micro mechanism in VOC 15-1:
>
>     |$\mathit{K}$|base|novel|all|
>     |:-:|:-:|:-:|:-:|
>     |1|78.6|30.9|67.2|
>     |3|79.7|34.0|68.8|
>     |5|80.1|36.8|**69.8**|
>     |7|80.4|32.8|69.1|
>
>     So we choose $\mathit{K}$ = 5 in VOC dataset.
>     The choice of $\mathit{K}$ = 1 in ADE20K is due to that ADE20K is an admittedly difficult dataset, which makes the pseudo-label generated by the old model inaccurate. Many historical classes would be mis-classified into the future class regions. Choosing a larger $K$ will exacerbate the misleading of inaccurate pseudo-label: pixels belonging to the same historical class will be forced to be separated into different categories. As a result, it confuses the model learning. Under the condition, we set $\mathit{K}$ = 1 for better performance.
>
> 3. **[How the performance varies for different class orders on ADE20K]**
> This is indeed an interesting idea. In fact, we have conducted similar experiments on VOC dataset with 20 different orders in our paper, as shown in Fig. 3 \(c\). However, due to time constraints, we are not able to provide results for swapping training orders on ADE20K (as it requires 40 days running on single NVIDIA GeForce RTX 3090 GPU for 20 different orders), but this could be an interesting future work.
>
> ### *Responses to questions*
> 1. **[Description about disjoint setting]**
> Sorry, this is just a description error. We will fix it in the revised version.
> 2. **[Are the joint results on ADE20k based on the proposed architecture, including the Mask2Former or for the base DeepLabv3 model?]**
> The joint results on ADE20k are based on the proposed model using proposals. The reason for the identical/low results is that we only adapt Mask2Former as a frozen pretrained proposal generator.   It does not fine-tuned or fit ADE20K dataset. Besides, this precisely illustrates that MicroSeg actually addresses the problem of incremental learning, rather than improving CISS results by boosting the performance of the basic semantic segmentation.
> 3. **[Details of pretrained ResNet101]**
> The pretrained weight we used for the ResNet-101 network was acquired from the PyTorch official site by using the torchvision package. Please refer to L10 in the supplementary material.
> 4. **[How does the proposed split of segment proposals and classification differ from [8]? And is the splitting of the task into segment proposal and classification novel for semantic segmentation or for CISS?]**
> Segment proposals are generated by proposal generator, which is identical to Mask2Former. While proposal classification is not related to Mask2Former. To the best of our knowledge, this method is novel in CISS.
> 5. **[Performance difference between classes present and absent in MS-COCO]**
> Yes, it differs between classes present and absent in MS-COCO. Although it is a bit infeasible to directly evaluate the segment proposal, we designed the following experiment to show the difference. We counted the classes that were shared in MS-COCO and ADE20K (20 in our experiment), to observe the variation of the mIoU performance of these classes with/without proposal classification branch on ADE20K dataset. The experiments are conducted on ADE 50-50.
>
>     ||shared|non-shared|all|
>     |:--:|:-:|:--:|:---:|
>     |w/o proposal|34.9|30.2|30.7|
>     |with proposal|38.0 (+3.1)|32.1 (+1.9)|32.9 (+2.2)|
>
>     Compared to non-shared classes, shared classes have better performance improvement with proposal classification branch. It also can be observed that the performance of both shared and non-shared has been improved, which shows the robustness of our method, even on the unseen classes to proposal generator.
> 6. **[Details of the contrastive loss]**
> Yes, contrastive loss is only applied for the background. It is used for the background after the pseudo-labeling, i.e. $c_{\hat{u}}$.
> 8. **[Which losses are applied to the baseline in Tab 3?]**
> Only the BCE loss was applied to the baseline. The better performance comes from the applied *freeze strategy*, described in Line 208. This was also validated in SSUL.

---

> > ### Comment · Reviewer_uMp6 · 2022-08-09
> > **Feedback on responses**
> >
> > The authors respond well to most questions and the new ablation adds to the substance of the paper.
> >
> > The fact that K=1 works best for ADE seems to indicate a problem with the formulation of the micro mechanism, although ADE is a bit special and one might have not much use of large Ks.

---

### Official Review · Reviewer_FY3k · 2022-07-09

**Rating:** 7
**Confidence:** 5
**Soundness:** 3 good
**Presentation:** 3 good
**Contribution:** 3 good

**Summary:**

This work address the "semantic shift of the background class" problem where some concepts learned at previous stages are wrongly assigned to the background class at the current training stage, in Class incremental semantic segmentation task. The authors first splits the given image into hundreds of segment proposals with a proposal generator pretrained on MSCOCO. Then the segment proposals with strong objectness from the background are clustered and assigned newly-defined labels during the optimization. Combining such supervision into a conventional dense prediction framework, the proposed method could relieve the catastrophic forgetting caused by the semantic shift of the background class accordingly. Experiments on Pascal VOC and ADE20K datasets show competitive results with state-of-the-art.

**Questions:**

Please see the weaknesses.

------------------------------------------
The authors have addressed my major concerns. I would stick to my original score.

**Ethics Review Area:**

["I don’t know"]

**Limitations:**

The authors adequately addressed the limitations and potential negative societal impact of their work.

**Strengths And Weaknesses:**

Strength:

1. The motivation is clearly presented and the method is straightforward to address the "semantic shift of the background class" problem where some concepts learned at previous stages are wrongly assigned to the background class at the current training stage, in Class incremental semantic segmentation task.

2. The paper is well written and easy to follow.

3. The overall performance of the proposed method is good and sufficient to verify the effectiveness of the proposed method.


Weakness:

1. The claim in Line 175~176 is not validated which it is valuable to see whether the proposed method could prevents potential classes from being incorrectly classified into historical classed.

2. In Tab. 1, for VOC 2-2 (10 tasks) and VOC 19-1 (2 tasks) MicroSeg gets inferior performance compared with SSUL, the reason should be explained. It also appear in ADE 100-50 (2 tasks) in Tab. 2.

3. The proposed method adopts a proposal generator pretrained on MSCOCO which aggregates more information. Is it fair to compared with other methods? Besides, could the proposed technique propmotes existing Class incremental semantic segmentation methods.

---

> ### Author Response · Authors · 2022-08-02
> **Response to Reviewer FY3k**
>
> ### *Responses to weaknesses*
> 1. **[The claim in Line 175~176 is not validated]**
> In the Micro mechanism, we represent it as a set of K classes to better cache the feature diversity in those loose future classes. The statement in Line 175~176 can be proved with the following evidence:
> 	-  From the ablation studies on the Micro mechanism in Tab. 3, the performances of both base classes and novel classes are improved with Micro mechanism(+9.9% in total, comparing the experimental setting of with and without Micro mechanism of Tab. 3).
> 	- Focused on historical classes, we further measure the probability of being incorrectly assigned to a historical class by: $$Err = 1 - Precision = \frac{FalsePositive}{TruePositive+FalsePositive}$$ We have completed experiments with and without the Micro mechanism in VOC 15-1, step 2, to observe the variation of $Err$. As a result, the historical classes have a mean $Err$ of 9.2% and 13.5%, for methods with and without the Micro mechanism, respectively. It indicates that the Micro mechanism does help to reduce error rate of being classified into historical classes.
> 	- This phenomenon can also be observed from the comparison of the qualitative analysis in Fig. 4. Taking the {*TV*} set (row 2 & 4 of Fig. 4, referring to SSUL and MicroSeg respectively) as an example, the potential future class (*TV*) is not incorrectly classified into the past class by MicroSeg, obviously superior to the result of SSUL.
> 2. **[MicroSeg gets inferior performance compared with SSUL (Tab. 1 & 2), the reason should be explained]**
> MicroSeg achieves better performance than SOTA method (SSUL) in most (4 of 6 scenarios in VOC, 3 of 4 scenarios in ADE20K) incremental scenarios. The reason why the other experimental results of MicroSeg in VOC are slightly lower than those of SSUL may be due to the small number of categories in some steps (e.g., step 2 of 19-1 has only 1 category). The lack of diverse samples makes model trained not well. ADE20K is an admittedly difficult dataset. The performance of MicroSeg on ADE 100-50 does have a trivial drop (-0.5% to SSUL). It is also worth pointing out that SSUL does not outperform all previous methods in ADE20K, e.g. ADE 50-50, compared with PLOP. It also shows that the task of CISS is still hard, especially on more challenging datasets like ADE20K. Thus, CISS research is valuable. We will add the analysis to the revised version.
>
> 3. **[Is it fair to compare with other methods? Besides, could the proposed technique promote existing Class incremental semantic segmentation methods?]**
> We have conducted a related experiment (L27-44) in Table 3 of the Supplementary Material to illustrate the fairness of the comparison, by replacing the saliency detector of SSUL with Mask2Former. Here are experimental details: we compared the performance of the original SSUL (1st row) and SSUL with the proposal generated by Mask2Former (2nd row) (i.e., *SSUL+mask proposal*). mask proposal for SSUL is obtained by binarization of the mean of queries in Mask2Former, which can measure the attention of all queries to a region, and we suppose that it is reasonable to use it as a saliency map. All experiments are done in VOC 15-1.
> - As a result, SSUL with Mask2Former obtains mIoU of 67.5, while MicroSeg achieves 69.8. Our proposed method performs better than SSUL, though SSUL also adopts Mask2Former.
> - Besides, the SSUL with Mask2Former achieves a comparable performance to the original SSUL (67.5 for SSUL + Mask2Former v.s. 67.6 for the original SSUL).
> - The result shows that the performance gain of MicroSeg was not mainly derived from the positive impact of Mask2Former on semantic segmentation (otherwise, applying it to SSUL should also result in a significant improvement). The key of our proposed method is proposals with the Micro mechanism, trying to address the problem of background shift in incremental learning.
>
>     As the application of segment proposal was not considered in previous CISS methods, the proposed method may not be able to naturally applied to them to address CISS. However, MicroSeg, serving as a simple baseline, provides an idea of mining unseen classes in the CISS task. We believe it will inspire and motivate following-up research in this direction.

---

### Official Review · Reviewer_Kd2M · 2022-07-10

**Rating:** 4
**Confidence:** 4
**Soundness:** 2 fair
**Presentation:** 3 good
**Contribution:** 2 fair

**Summary:**

The paper proposes MicroSeg for class incremental semantic segmentation. The region proposal generator is used to detect the potential objects with strong objectness in the background. The these identified proposals are grouped into a new defined label during the optimization. Experiments are performed on both Pascal VOC and ADE20K to validate the idea.

**Questions:**

I will consider to raise the rating if my concerns in the above weakness section can be well addressed.

**Limitations:**

Yes.

**Strengths And Weaknesses:**

**Strengths**:

1.The paper has a good motivation, where incremental semantic segmentation is a meaningful to explore and the discussed background shift problem is also difficult for existing models.

2.The experiments show an obvious improvement to existing methods. Table 3 shows the effect of Proposal and Micro respectively.

3.The paper is organized well with good writing and figures.

**Weakness**:

1.The tech contribution of MicroSeg is very limited. Region proposal network for class-agnostic detecting novel objects is already widely used, such as [a].

2.SSUL uses the off-the-shelf saliency-map detector to detect unseen classes while the paper uses the pretrained Mask2Former to produce the region proposals. This may introduce an unfair comparison in both data and model size. Mask2former is additionally trained on COCO and has much larger parameters than the off-the-shelf detector. What if SSUL also adopts Mask2Former to detect unseen classes? Or what if SSUL can generate the object proposals in an unsupervised way w/o additional lots of data or heavy model, such as [b].

3.Missing experiments of using other region proposals networks instead of using Mask2Former, such as the RPN in Mask R-CNN? Will it influence the final model performance a lot?

4.Missing the speed comparison and model parameters with other methods. What are the model sizes of the proposed MicroSeg combined with Mask2Former?

[a] Gu, Xiuye, et al. "Open-vocabulary object detection via vision and language knowledge distillation. ICLR, 2022.

[b] Open-World Instance Segmentation: Exploiting Pseudo Ground Truth From Learned Pairwise Affinity. CVPR, 2022.

---

> ### Author Response · Authors · 2022-08-02
> **Response to Reviewer Kd2M**
>
> ### *Responses to weaknesses*
> 1. **[Limited contribution of MicroSeg. Region proposal network (RPN) is already widely used, such as [a]]**
>
> - First, [a] studied the open-set object detection task, while our work focuses on semantic segmentation in the class incremental learning (CISS) setting. The proposal generator in MicroSeg is different to RPN in object detection, since our method requires class-agnostic masks while RPN only can produce bounding boxes. We will cite the suggested paper and revise the description of our proposal generator part.
> - Second, MicroSeg aims to accomplish the CISS with regional objectness information rather than the mask proposal generator.
> - Finally, we use proposals not as a simple appropriation, but with careful consideration of the characteristics of the CISS task,  in which there are potential categories in the background, and these categories can be mined by proposals. To the best of our knowledge, MicroSeg is the first application of regional objectness to CISS.
>
> 2. **[Off-the-shelf saliency detector for SSUL compared to Mask2Former in the paper]**
> Thanks for the suggestions. We have conducted a related experiment (L27-44) in Table 3 of the Supplementary Material to illustrate the fairness of the comparison, by replacing the saliency detector of SSUL with Mask2Former. Here are experimental details: we compared the performance of the original SSUL (1st row) and SSUL with the proposal generated by Mask2Former (2nd row) (i.e., *SSUL+mask proposal*). mask proposal for SSUL is obtained by binarization of the mean of queries in Mask2Former, which can measure the attention of all queries to a region, and we suppose that it is reasonable to use it as a saliency map. All experiments are done in VOC 15-1.
> - As a result, SSUL with Mask2Former obtains mIoU of 67.5, while MicroSeg achieves 69.8. Our proposed method performs better than SSUL, though SSUL also adopts Mask2Former.
> - Besides, the SSUL with Mask2Former achieves a comparable performance to the original SSUL (67.5 for SSUL + Mask2Former v.s. 67.6 for the original SSUL).
> - The result shows that the performance gain of MicroSeg was not mainly derived from the positive impact of Mask2Former on semantic segmentation (otherwise, applying it to SSUL should also result in a significant improvement). The key of our proposed method is proposals with the Micro mechanism, trying to address the problem of background shift in incremental learning.
>
>     For the other suggested experiment about GGN[b], we use their official pretrined model as a saliency detector: we extract masks after PA and Grouping of GGN, and choose top 20 objects to generate saliency maps. Here is the comparison with original SSUL on VOC 15-1. It can be observed that there is a performance drop for SSUL when using [b].
>
>     ||base|novel|all|
>     |:--:|:--:|:--:|:--:|
>     |SSUL|77.3|36.6 |67.6|
>     | SSUL + [b]|75.1|26.5|63.5|
>
>     We would be happy to conduct additional experiments to illustrate the fairness of the comparison, if there are any other possible suggested settings.
>
> 3. **[Experiments of using other region proposals networks such as RPN in Mask R-CNN]**
>     There might be a misunderstanding. Proposals generated by RPN cannot be directly applied to our MicroSeg, since our method requires class-agnostic masks while RPN can only produce bounding boxes. To address the concerns, we conduct additional experiments using other types of mask proposals. Specifically, we generate proposals with MaskFormer and Mask2Former. We also explore the number of proposals $\mathit{N}$.
>
>     ||base|novel|all|#params.|
>     |:--:|:-:|:-:|:-:|:-:|
>     | MaskFormer($\mathit{N}$=100)|79.3|34.1|68.5|44M|
>     | Mask2Former($\mathit{N}$=100)|80.1|36.8|69.8|107M|
>     | Mask2Former($\mathit{N}$=200)|80.4|37.3|70.1|216M|
>
>     It can be observed that the quality of the proposal affects the experimental results to some extent, but MicroSeg still achieves SOTA performance, even with MaskFormer($\mathit{N}$=100) (68.5 v.s. 67.6, compared to SSUL, in VOC 15-1).
>
> 4. **[Missing speed comparison and model parameters with other methods]**
> As suggested, we provide a comparison of the performance (mIoU) of VOC 15-1, computational complexity (GFLOPS) and model size (#params) of the three methods.
>
>     ||mIoU|GFLOPS|#params.|
>     |:-:|:--:|:-:|:-:|
>     |PLOP| 54.6 |677|58M|
>     |SSUL| 67.6 |211|60M|
>     |MicroSeg (Ours)|**69.8**|234|65M|
>
>     MicroSeg achieves performance gains of 2.2% at similar model sizes and GFLOPS. In addition, the model size of Mask2Former that we use for proposal generator is 107M. However, it is frozen to generate proposals without training. Note, even with a smaller scale MaskFormer (44M of model size), our proposed MicroSeg still achieves SOTA performance(68.5 for MicroSeg v.s. 67.6 for SSUL, in VOC 15-1).

---

> > ### Comment · Reviewer_Kd2M · 2022-08-03
> > **The usage of RPN combining with class-agnostic mask head**
> >
> > Thanks for the response. I give one quick comment here. It's true that RPN can only output bounding boxes, but it can output masks when combing with the class-agnostic mask head. Please refer to Figure 4 (mask visualization) and implementation details of [a].

---

> > > ### Author Response · Authors · 2022-08-03
> > > **Thanks for the suggestions of experiments**
> > >
> > > Thanks for the quick response. We will follow your suggestions and provide the corresponding results here as soon as possible.

---

> > > > ### Author Response · Authors · 2022-08-05
> > > > **Response to Reviewer Kd2M**
> > > >
> > > > Following your kind suggestions, we have finished the experiments of generating proposals by RPN in Mask R-CNN combined with class-agnostic segmentation head (denote as RPN+Seghead). As shown in the following table,  using RPN+Seghead as a proposal generator does not significantly affect the performance, comparing to the original setting of MicroSeg. We conduct the experiment on VOC 15-1. We will include the result into the revised version and hope it can help address your concerns in weakness #3.
> > > >
> > > > ||base|novel|all|
> > > > |:--:|:-:|:-:|:-:|
> > > > | RPN+Seghead|80.5|33.0|69.2|
> > > > | MicroSeg|80.1|36.8|69.8|

---

### Official Review · Reviewer_rFyJ · 2022-07-11

**Rating:** 7
**Confidence:** 4
**Soundness:** 3 good
**Presentation:** 4 excellent
**Contribution:** 3 good

**Summary:**

This paper proposes a novel framework called MicroSeg for the catastrophic forgetting problem caused by the semantic shift of the background class. This paper pioneered the idea of considering unknown categories that may occur in the future in advance in background learning. More specifically, the idea is quite interesting with the assumption that background regions with strong objectness possibly belong to those concepts in the historical or future stages.  Experiment results show promising results on Pascal VOC and ADE20K datasets.

**Questions:**

1. In the Label Remodeling part (i.e., Equation 3), what is the meaning of symbols, such as ^ and \ ?
2. The Proposal Generator has the ability to generate a binary map. How is it trained? Is it a pretrained model from [8]?

**Limitations:**

The authors have well addressed the limitations.

**Strengths And Weaknesses:**

Strengths
1. Overall the paper is well written, well illustrated, and subsequently pleasant to read.
2. The method itself is clear and proposed contributions are really intuitive.
3. The contributions and experiments are decent and clearly shows improvement over the state-of-the-art. Ablations are okay.

Weakness
1. The author proposes pre-learning for future categories, so the labeling of unknown categories that may appear in the future will be the key to this paper. However, Label Remodeling in Section 3.3 is not explained clearly. See the the Questions part for details.
2. There are few descriptions of MicroSeg-M in the paper. It just shows that it uses the memory sampling strategy [5]. In my opinion, simply migrating other methods to this task may not be a contribution. If authors have no special design on MicroSeg-M, I suggest to remove it, since MicroSeg itself is interesting enough.
3. There are several formatting errors in the article. For instance, there are no full stops on lines 42 and 66. The authors are desired to read the paper thoroughly and revise them carefully.

---

> ### Author Response · Authors · 2022-08-02
> **Response to Reviewer rFyJ**
>
> ### *Responses to weaknesses*
> 1. **[Meaning of symbols in Label Remodeling (Sec. 3.3)]**
> The symbol `∧` represents the co-taking of conditions, i.e. the satisfaction of two conditions. That is, both the conditions of (1) being labeled as background in ground truth and (2) having high confidence in the predictions of the $f_{t-1,\theta}$ belonging to the past category, are satisfied. The symbol `\` means the difference of the set, i.e. the part of the background that does not belong to the past categories.
> 2. **[Description of MiscroSeg-M (the memory sampling strategy [5])]**
> Thanks for the suggestions. The version of MicroSeg with memory (MicroSeg-M) was mainly presented for a fair comparison with SSUL-M. We will revise the paper to make it more clear.
> 3. **[Formatting issues (missing full stops)]**
> We appreciate the reviewer pointing this out, and will carefully and thoroughly check and revise the paper.
>
> ### *Responses to questions*
> 1. **[Label Remodeling symbols]**
> Please refer to the response to the above weakness #1.
> 2. **[How is the Proposal Generator trained? pretrained model from [8]?]**
> Yes, we use the off-the-shelf Mask2Former pre-trained model to generate the proposals.

---

### Comment · Area_Chair_C3Gs · 2022-08-06
**Discussion**


Dear Reviewers,

Thank you for reviewing this paper. Since authors have submitted their rebuttal, please check if the rebuttal addresses your concerns. Please also check the comments from other reviewers, and if you have any question, please discuss with authors in OpenReview soon.

Best Regards,

AC

---

### Author Response · Authors · 2022-08-07
**To Reviewers**

Dear Reviewers,

Thanks for the comments and suggestions for this paper. We have tried our best to address all concerns. We would appreciate receiving your feedbacks, as the discussion deadline is approaching. If you have additional questions, we would be happy to reply.

Sincerely yours,

Authors

---

### Meta-Review · Area_Chair_C3Gs · 2022-08-24

**Recommendation:** Accept
**Confidence:** Less certain

**Metareview:**

Most of the reviewers pointed out that the motivation of the method is clear, and the method is novel and interesting. The proposed method is also effective on multiple benchmarks. One of the reviewer has concerns about the choice of a parameter (K), and another reviewer has concerns about details of the method. AC admits that these points need to be further improved, but thinks these points can be clarified in the camera ready version. Thus, considering the novelty and efficacy of the method, AC recommends accept for this paper, yet suggests the authors to carefully take the reviewers' comments into account when preparing the final version.

**Award:**

No

---

### Decision · Program_Chairs · 2022-09-14

Accept